# An upgraded high-precision gridded precipitation dataset for the Chinese mainland considering spatial autocorrelation and covariates

Jinlong Hu[1], Chiyuan Miao[1, *], Jiajia Su[1], Qi Zhang[1], Jiaojiao Gou[1, 2], Qiaohong Sun[3, 4]

1 State Key Laboratory of Earth Surface Processes and Disaster Risk Reduction, Faculty of Geographical Science, Beijing Normal University, Beijing 100875, China
2 Department of Geographic Science, Faculty of Arts and Sciences, Beijing Normal University, Zhuhai 519087, China
3 State Key Laboratory of Climate System Prediction and Risk Management/Key Laboratory of Meteorological Disaster, Ministry of Education/Collaborative Innovation Center on Forecast and Evaluation of Meteorological Disasters, Nanjing University of Information Science and Technology, Nanjing 210044, China
4 School of Atmospheric Sciences, Nanjing University of Information Science and Technology, Nanjing 210044, China

*Correspondence to*: Chiyuan Miao (miaocy@bnu.edu.cn)

**Abstract.** Precipitation is a critical driver of the water cycle, profoundly influencing water resources, agricultural productivity, and natural disasters. However, existing gridded precipitation datasets exhibit markable deficiencies in capturing the spatial autocorrelation and associated environmental and climatic influences—here referred to collectively as precipitation-related covariates—which limits their accuracy, particularly in regions with sparse meteorological stations. To address these challenges, this study proposes a completely new gridded precipitation generation scheme that integrates long-term daily observations from 3,746 gauges with 11 key precipitation-related covariates. Building upon the improved inverse distance weighting interpolation method used in our previous dataset CHM_PRE V1, we integrated a machine learning algorithm—light gradient boosting machine (LGBM)—to incorporate precipitation-related covariates in a data-driven manner. This integration allows for a more comprehensive characterization of precipitation patterns, jointly capturing spatial autocorrelation and covariate-based variability. By this novel scheme, a new high-precision, long-term, daily gridded precipitation dataset for the Chinese mainland (CHM_PRE V2) was developed. Validation against 63,397 high-density gauges demonstrated that CHM_PRE V2 significantly outperforms existing gridded precipitation datasets. Specifically, it achieves a mean absolute error of 1.48 mm/day and a Kling-Gupta efficiency of 0.88, representing improvements of 12.84% and 12.86%, respectively, compared to the previously optimal dataset. Regarding precipitation event detection, CHM_PRE V2 achieved a Heidke skill score of 0.68 and a false alarm ratio of 0.24, surpassing the previously optimal dataset by 17.24% and 29.17%, respectively. These results demonstrate that CHM_PRE V2 markedly enhances precipitation measurement accuracy, reduces overestimation of precipitation events, and provides a reliable foundation for hydrological modelling and climate assessments. This dataset features a resolution of 0.1°, spans from 1960 to 2023, and will be updated annually. Free access to the dataset can be found at https://doi.org/10.5281/zenodo.14632156 (Hu and Miao, 2025).

# 1 Introduction

Precipitation serves as the pivotal factor driving the water cycle, directly influencing the distribution and variability of water resources, agricultural productivity, ecosystem health, and the occurrence and progression of natural disasters (Ham et al., 2023; Sun et al., 2018; Zhang et al., 2017; Zheng et al., 2025) At regional and global scales, gridded precipitation datasets provide detailed spatial resolution and temporal continuity, making them fundamental in hydrological and climate sciences and disaster forecasting (Qiu et al., 2024; Sun et al., 2021; Tang et al., 2021; Xiong et al., 2024). However, due to the high spatiotemporal variability of precipitation and the complexity of observation conditions, generating high-precision gridded precipitation data remains a formidable challenge (Jiang et al., 2023).

In China, various types of precipitation datasets have been extensively utilized in research, encompassing products derived from data assimilation techniques, remote sensing techniques, and gauge-based interpolation techniques. Precipitation data derived from data assimilation (Gelaro et al., 2017; Hersbach et al., 2020; Rodell et al., 2004) integrate meteorological models with observational data to provide highly consistent datasets. However, their accuracy is often constrained by the physical parameterization schemes of the models. Remote sensing-based precipitation datasets (Ashouri et al., 2015; Huffman et al., 2007, 2015; Kubota et al., 2020) offer global or regional precipitation distributions via satellite observations, ensuring extensive spatial coverage. Nonetheless, their precision is limited by data resolution and satellite orbital constraints, particularly in regions with complex terrain and high latitudes. Precipitation gauges, as the most direct and accurate tools for measuring precipitation, allow for gridded precipitation datasets generated through interpolation, effectively capturing the localized characteristics of precipitation with high accuracy (Harris et al., 2020; He et al., 2020; Qin et al., 2022; Shen et al., 2010; Wu and Gao, 2013; Xie et al., 2007).

Our previous study developed a gridded precipitation dataset for the Chinese mainland (a member of the China Hydro-Meteorology datasets, hereinafter called CHM_PRE V1) based on inverse-distance weighting interpolation method and parameter-elevation regression on independent slopes model (PRISM) (Daly et al., 1994, 2002), using data from 2,839 gauges. The CHM_PRE V1 demonstrates overall high accuracy across the Chinese mainland (Han et al., 2023), and has received widespread attention and extensive use, benefiting a large number of hydro-meteorological related studies (Hu et al., 2024; Wan and Zhou, 2024; Yin et al., 2025). However, interpolation-based precipitation datasets rely heavily on ground meteorological gauges, performing poorly in areas with sparse station distribution or missing data.

In summary, a key limitation of existing datasets is that they tend to focus on either spatial autocorrelation or a limited set of precipitation-related covariates, but rarely incorporate multiple types of information simultaneously. However, precipitation is influenced not only by spatial autocorrelation—that is, the dependence of precipitation at a given location on surrounding areas (Chen et al., 2010, 2016; Fan et al., 2021; Huff and Shipp, 1969; Tang et al., 2020)—but also by a wide array of covariates, such as elevation, land surface conditions, atmospheric parameters, and recent precipitation events (Adler et al., 2008; Ham et al., 2023; Ravuri et al., 2021; Trucco et al., 2023). This lack of comprehensive consideration for multiple covariates constrains the accuracy of these datasets, particularly in regions with sparse meteorological stations, such as

western China (Jiang et al., 2023). Moreover, existing methods tend to generate excessive minor precipitation, leading to an overestimation of precipitation events, which will have considerable impacts on hydrologic modelling (Dong et al., 2020; Kang et al., 2024; Wei et al., 2022).

To address the aforementioned issues, this study introduces a new high-precision, long-term daily gridded precipitation dataset for the Chinese mainland (a member of the China Hydro-Meteorology datasets, hereinafter called CHM_PRE V2). Building on CHM_PRE V1, CHM_PRE V2 integrates precipitation gauges, remote sensing observations, reanalysis data, and various precipitation-related factors. Through the use of advanced spatial interpolation and machine learning algorithms, our method captures spatial autocorrelation while jointly modelling multiple covariates to enhance precipitation accuracy. As a result, we obtain a high-accuracy gridded dataset that covers the entire Chinese mainland (18°N–54°N, 72°E–136°E). The spatial resolution of the dataset is set to 0.1° to maintain consistency with our previous dataset (Han et al., 2023; Zhang et al., 2025). The dataset spans the period from 1960 to 2023 and will be updated annually. CHM_PRE V2 not only enhances the accuracy of precipitation measurements but also significantly reduces overestimations of precipitation events. The high-precision gridded precipitation dataset can reduce the uncertainty in hydrological modelling and analysis, providing a more reliable data foundation for hydrologic and climatological studies. For clarity, a list of abbreviations used throughout this paper is presented in **Table S1** in the supplementary materials.

## 2 Data

The CHM_PRE V2 dataset was developed using extensive precipitation gauge observations, supplemented with a diverse array of ancillary datasets that serve as precipitation covariates. These covariates include satellite-derived products, land surface model outputs, and various geophysical and meteorological variables, aiming to enhance the characterization of precipitation, particularly in regions with sparse observational coverage. This integration of multi-source information is designed to improve the spatial continuity and accuracy of the precipitation estimates across the Chinese mainland. **Figure 1** illustrates details of the various datasets utilized in CHM_PRE V2 construction, including dataset names, original spatial and temporal resolutions, and coverage periods. In total, 16 datasets from 11 distinct categories were incorporated. These datasets collectively provide critical information on land surface properties, atmospheric conditions, and recent precipitation patterns that influence precipitation generation and distribution. In addition, the CHM_PRE V2 dataset is designed to represent precipitation characteristics across the Chinese mainland, excluding Taiwan, Hong Kong, Macau, and other Chinese islands. In the following sections, we will provide a detailed introduction to the data sources employed in the construction of the CHM_PRE V2 dataset.

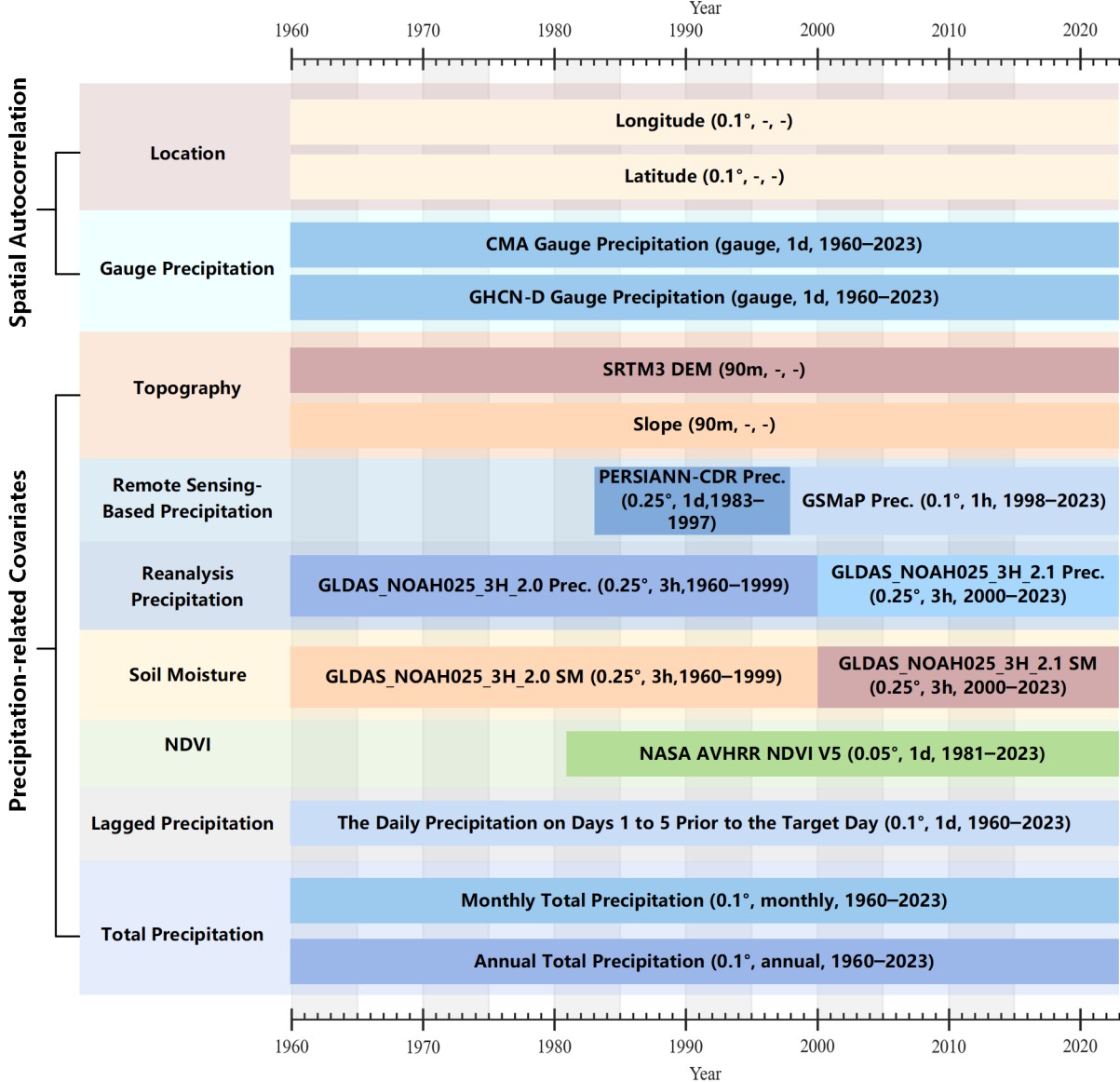

**Figure 1. The data used for precipitation retrieval.**

## 2.1 Spatial autocorrelation data

CHM_PRE V2 incorporates comprehensive daily precipitation gauge data to support spatial autocorrelation modelling. The primary daily precipitation gauge data sourced from the China Meteorological Administration (CMA; http://data.cma.cn, last access: January 2024) spans the entire Chinese mainland, encompassing records from 2,816 stations between 1960 and 2023. Daily precipitation is defined as the cumulative precipitation recorded between 20:00 on one day and 20:00 on the following

day (local time in Beijing), with all data subjected to rigorous quality control (Zhang et al., 2020). To mitigate the limit of boundary effects (Ahrens, 2006), additional precipitation gauges near the Chinese mainland were obtained from the Global Historical Climatology Network-Daily Version 3 (GHCND) dataset. The GHCND is a reliable and globally comprehensive climate dataset, and maintained by the National Climatic Data Center (NCDC) of the National Oceanic and Atmospheric Administration (NOAA) (Durre et al., 2008, 2010; Menne et al., 2012). The GHCND dataset was sourced from NOAA (https://www.ncei.noaa.gov/products/land-based-station/global-historical-climatology-network-daily) on September 11, 2024. To ensure data quality, only stations with more than 70% effective days (over 255 days) in a year were retained for dataset construction. **Figure 2**(a) illustrates the spatial distribution of both CMA and GHCND stations, while **Figure 2**(b) shows their annual availability. Over time, the number of available CMA stations increased from 1,992 in 1960 to 2,767 in 2023, improving spatial coverage considerably. In contrast, the number of accessible GHCND stations in the region declined from 674 in 1960 to 264 in 2023.

## 2.2 Precipitation-related covariate data

The Shuttle Radar Topography Mission (SRTM) Digital Elevation Model (DEM) dataset was utilized to characterize the influence of elevation on precipitation and to generate slope data. In this study, we used the SRTM DEM V4 acquired from the Consortium for Spatial Information, Consultative Group for International Agricultural Research (CGIAR-CSI, https://srtm.csi.cgiar.org/) on August 8, 2024, with a spatial resolution of 3 arc-seconds (approximately 90 meters near the equator). The SRTM DEM V4 was generated based on National Aeronautics and Space Administration (NASA) SRTM DEM V1, and has undergone post-processing of the NASA data to "fill in" the no data voids, such as water bodies (lakes and rivers), areas with snow cover and in mountainous regions (e.g., the Himalayas), resulting in seamless elevation for the globe. To enhance the spatial and temporal detail of precipitation estimation, two satellite-based precipitation products—the Global Satellite Mapping of Precipitation (GSMaP) and the Precipitation Estimation from Remotely Sensed Information using Artificial Neural Networks (PERSIANN-CDR) dataset—were incorporated as covariates. GSMaP V8 data spans from 1998 to the present with 0.1° spatial and 1-hour temporal resolution (Kubota et al., 2020). We acquired the GSMaP data from Japan Aerospace Exploration Agency (JAXA; https://sharaku.eorc.jaxa.jp) on September 9, 2024, and used the data from 1998 to 2023. PERSIANN-CDR data spans from 1983 to the present (Ashouri et al., 2015), and the data from 1983 to 1997 was used for the retrieval.

The precipitation and soil moisture from the Global Land Data Assimilation System Noah Land Surface Model (GLDAS NOAH) (Rodell et al., 2004) were also used for the retrieval. The data spans from 1960 to 1999 and the data spans from 2000 to 2023 were acquired from the GLDAS Noah L4 V2.0 and GLDAS Noah L4 V2.1 datasets. The NOAA Climate Data Record (CDR) of AVHRR Normalized Difference Vegetation Index (NDVI) (Vermote and NOAA CDR Program, 2019) was utilized to depict the vegetation characteristics, and the data from 1981 to 2023 was used.

In addition to spatial and environmental variables, precipitation temporal features were also introduced as covariates. Two types of temporal indicators were constructed: (1) the cumulative precipitation of the current month and year, representing

broader-scale precipitation conditions; and (2) daily lagged precipitation values from the previous five days, capturing short-term fluctuations. Each of these five recent days was treated as a separate variable. For example, the variable named "1st-day prior Prec." refers to precipitation one day before the current date, while "5th-day prior Prec." corresponds to five days prior.

## 2.3 Other datasets

To verify the reliability of the proposed CHM_PRE V2, we compared it with five existing gridded precipitation datasets. These datasets include GSMaP, PERSIANN-CDR, and GLDAS precipitation datasets, as mentioned above. Additionally, CHM_PRE V1 (Han et al., 2023), previously developed by our team, and the Integrated Multi-satellitE Retrievals for GPM (IMERG) Final L3 V7 precipitation dataset (Huffman et al., 2023) were also included in the comparison. Details of the original spatiotemporal resolution and accessible time span of these datasets are provided in **Table S2** in the supplementary materials. All datasets were resampled to daily values at a resolution of 0.1°. To ensure a fair comparison, the analysis focused on the period from 2001 to 2022, during which all datasets were available.

To further validate the reliability of precipitation data, we obtained daily precipitation observations from 72,901 high-density automatic rain gauge stations across the Chinese mainland (hereafter we refer to it as CMA-HD), provided by the National Meteorological Information Center of CMA (Li et al., 2018). The data spans the period from 2013 to 2019, and we got 63,397 available stations after quality control and annual integrity control. **Figure 2**(c) illustrates the number of CMA-HD stations within each 0.1° grid cell. The dataset demonstrates high station density throughout the eastern region, while maintaining basic coverage in the northwest and Tibetan Plateau areas. This extensive distribution ensures the validation results based on this dataset are highly reliable. Additionally, to examine the dataset's performance across various regions, we adopted the climatic regionalization scheme proposed by Ren et al.(1985), dividing China into seven distinct regions shown in **Figure 2**(d): North East China (NEC), North China (NC), South and Central China (SCC), Inner Mongolia (IM), North West China (NWC), South West China (SWC) and Qinghai-Tibet Plateau (QT).

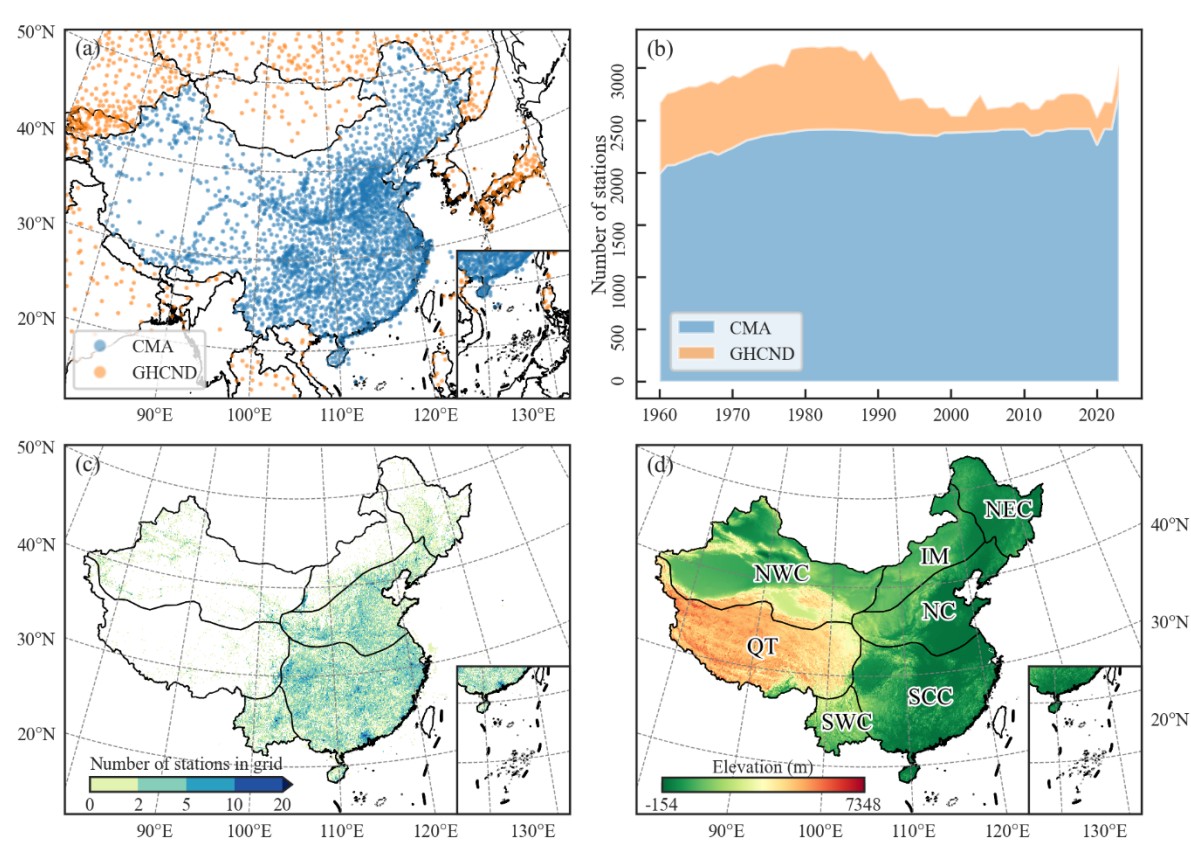


**Figure 2. (a) locations of CMA and GHCND stations used for precipitation retrieval; (b) the numbers of annual availability of precipitation stations; (c) locations of CMA-HD stations used for validation; (d) climatic regions.**

## 3 Methodology

The generation of CHM_PRE V2 can be divided into three stages: data preprocessing, precipitation interpolation based on
spatial autocorrelation, and precipitation retrieval based on covariates. **Figure 3** depicts the detailed steps involved in creating the dataset, which we will now introduce step by step.

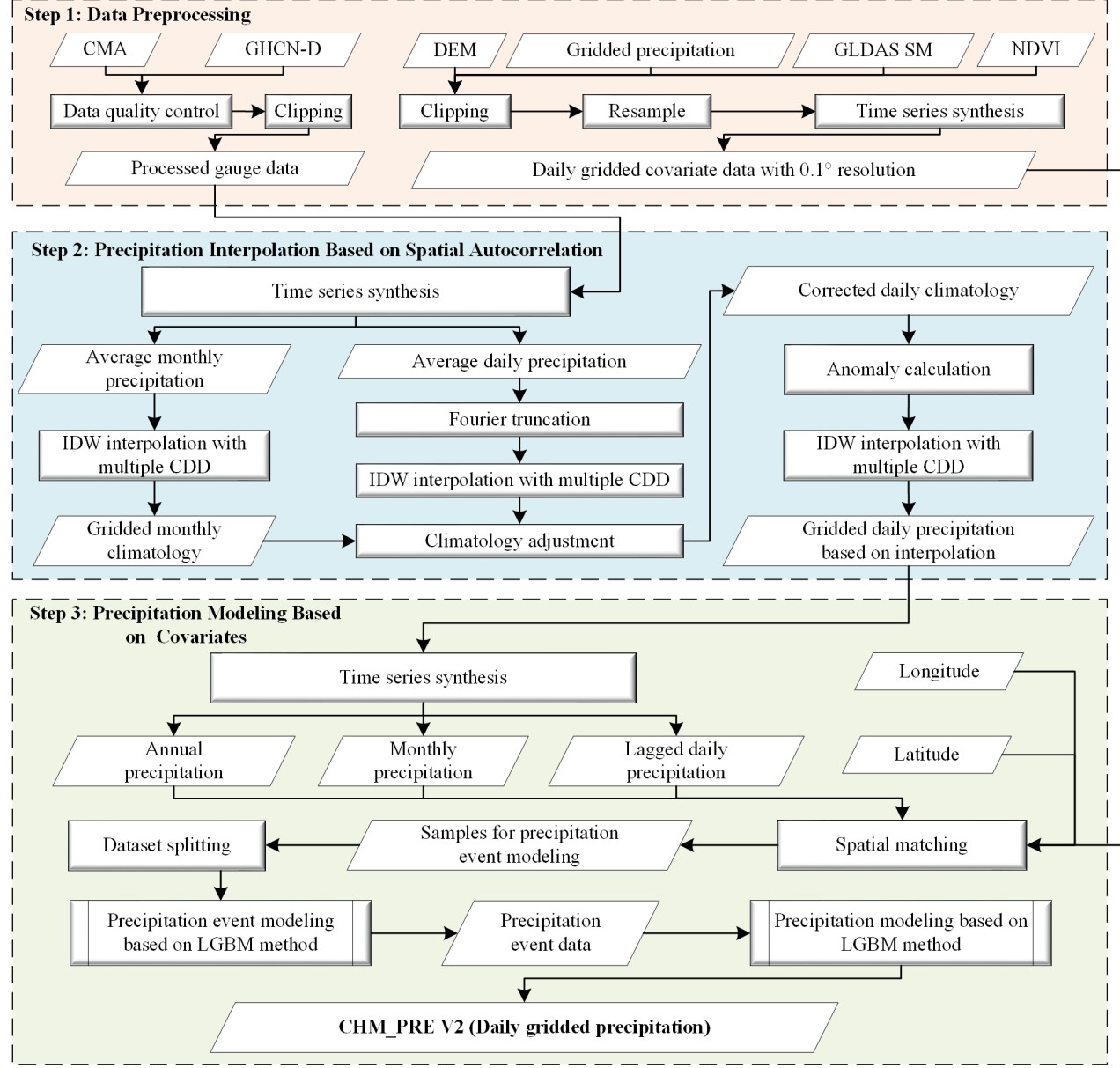

**Figure 3. The flowchart for dataset generation.**

## 3.1 Data Preprocessing

Data preprocessing consists of two main components: gauge data preprocessing and gridded data preprocessing. Initially, we performed quality control on the CMA and GHCND gauge data and excluded stations outside the region of interest. The latitude and longitude range of primary interest to us in the Chinese mainland spans approximately 18°N to 54°N and 72°E to 136°E. However, to mitigate boundary effects, we have extended the area of interest outward by roughly 3°, defining it as

15°N to 57°N and 70°E to 140°E. The remaining stations were merged to serve as the gauge dataset for retrieval. Similarly, for various gridded datasets, data outside the region of interest were removed, and all data were resampled to a spatial resolution of 0.1°. Finally, the gridded data were converted to a daily time scale, resulting in the final gridded covariate data for retrieval.

## 3.2 Precipitation interpolation based on spatial autocorrelation

Spatial autocorrelation is the most significant characteristic of precipitation data, and the most common approach to constructing gridded precipitation datasets involves interpolation based on gauge data (Harris et al., 2020; He et al., 2020). Consequently, in this section, we also utilize gauge-based interpolation to characterize the spatial autocorrelation of precipitation. The inverse distance weighting (IDW) method is widely used for interpolation due to its simplicity and computational efficiency. As a global interpolation method, IDW considers only the distance factor, applying inverse

distance weighting to all stations for interpolation. However, the spatial autocorrelation of many geographical features is often non-uniform. For example, many features may exhibit strong spatial autocorrelation within a specific distance range, which rapidly diminishes beyond that range. To address this, Shepard (1968) introduced the concept of correlation decay distance (CDD) into interpolation and proposed the adaptive distance weighting (ADW) method. CDD measures how the spatial correlation between stations decreases with increasing distance and ensures that the search radius is set to an

appropriate value, rather than using a fixed value for all situations (Dunn et al., 2020). Numerous datasets employ this method to interpolate gauge data to grids (Caesar et al., 2006; Dunn et al., 2020; Harris et al., 2020; Zhang et al., 2025). Based on this, Han et al. (2023) incorporated CDD into the IDW method, and calculated the CDD values suitable for interpolating precipitation over the Chinese mainland. Given a target grid cell $G$ surrounded by $n$ known stations $\boldsymbol{P}$ $\{P_1, P_2, \ldots, P_n\}$, where the precipitation value at station $P_i$ is $z_i$, the precipitation value at grid cell $G$ is calculated as:

$$z(G) = \frac{\sum_{i=1}^{n} d(G, P_i)^{-p} z_i}{\sum_{i=1}^{n} d(G, P_i)^{-p}} \tag{1}$$

where $d(G, P_i)$ represents the distance (km) between grid cell $G$ and gauge station $P_i$, and $p$ is the distance weighting exponent. In this study, $p$ is set to 2, representing the Euclidean distance.

The selection of the station set $\boldsymbol{P}$ for interpolation markedly impacts the interpolated results. In this study, we adopt the improved IDW method and use the CDD values calculated by Han et al. (2023) for interpolating precipitation over the

Chinese mainland (CDD1=244.7 km, CDD2=1336 km). When more than three stations are available within the CDD1 range, CDD1 is used as the interpolation CDD; otherwise, CDD2 is applied. Meanwhile, if more than ten stations are available within the interpolation CDD range, only the ten closest stations to the grid cell are used, to mitigate overestimation of precipitation events in the densely populated station areas of eastern China.

Furthermore, previous research has demonstrated that interpolated precipitation anomalies (Harris et al., 2020; He et al.,

2020) generally yield higher accuracy compared to direct precipitation interpolation. Thus, we adopt the interpolation scheme based on climatology anomaly rather than interpolating the raw precipitation values. To achieve this, daily and

monthly precipitation climatology data were generated. First, we calculated the average daily precipitation for 1971–2000 to derive preliminary station-level daily climatology data by using the daily precipitation gauges from the previous step. The daily climatology series at each station was then processed by the Fourier truncation, retaining only the first six harmonic components to suppress high-frequency noise (Xie et al., 2007). Subsequently, the station-level daily climatology data were interpolated using the improved IDW, producing preliminary gridded daily climatology data. To enhance the reliability of the daily climatology data, we followed the same procedure to generate gridded monthly precipitation climatology data. These monthly data were used to correct the gridded daily climatology, yielding the adjusted gridded daily climatology data (Han et al., 2023). The precipitation anomalies were defined as the difference between the actual station precipitation and the adjusted gridded daily climatology data. Finally, the station-level daily precipitation anomalies were interpolated using the improved IDW method. The gridded daily precipitation data based on interpolation were finally obtained by summing the interpolated anomalies with the adjusted gridded daily climatology data.

## 3.3 Precipitation retrieval based on covariates

Except spatial autocorrelation, precipitation is influenced by a range of meteorological factors that vary over space and time. However, most existing gridded precipitation datasets tend to model these aspects in isolation, often focusing solely on spatial autocorrelation or meteorological inputs, which may constrain the accuracy and generalizability of the datasets, especially in regions with sparse gauge coverage. To address this limitation, we propose a novel framework that integrates multiple precipitation covariates into a unified machine learning-based retrieval system, thereby enhancing the fidelity of precipitation estimates. To model spatial autocorrelation, we employed gridded precipitation data derived from gauge-based interpolation in Section 3.2, along with geographic coordinates (longitude and latitude). Precipitation covariates were drawn from various sources, including topographic features (elevation and slope), satellite-derived precipitation estimates, reanalysis-based precipitation products, soil moisture, and the normalized difference vegetation index (NDVI). Recent daily precipitation records and aggregate precipitation metrics were also incorporated to capture the temporal variability and underlying climatological patterns. The details of the retrieval data can be found in **Figure 1**.

To synthesize these spatial and covariate-based features, we employed a machine learning regression framework using the light gradient boosting machine (LGBM) algorithm. This model enables the flexible representation of complex nonlinear relationships between precipitation and its associated covariates, surpassing the limitations of conventional linear regression models. While linear regression models are the most commonly used response models, they are limited by their inability to capture nonlinear relationships and their relatively weak fitting capacity (Breiman, 2001; Chen and Guestrin, 2016; Yang et al., 2021). Machine learning-based models, in contrast, offer significant improvements in fitting performance and are more effective in representing nonlinear relationships (Guo et al., 2024; Hu et al., 2023). Among numerous machine learning-based models, LGBM, developed by Microsoft (Ke et al., 2017), is renowned for its high precision and high generalizability. Fundamentally, it employs a series of decision tree models for iterative training, progressively minimizing errors (or residuals) to ultimately generate predictions through a weighted summation. Unlike traditional gradient-boosted decision tree

(GBDT) methods, LGBM utilizes a histogram-based technique for data binning, rather than processing each individual data record. This method iterates, calculates gains, and splits data accordingly (Zhang and Gong, 2020). Gradient-based one-side sampling is employed to sample the dataset, assigning greater weights to data points with larger gradients during gain computation. Under equivalent sampling rates, this method often outperforms random sampling (Candido et al., 2021). Owing to these features, LGBM demonstrates exceptional accuracy and generalization, making it widely applicable to

various tasks such as classification, regression, and ranking (Bian et al., 2023; Jiang et al., 2024; Zhang et al., 2024). Hu et al. (2023) applied LGBM to the retrieval of suspended sediment concentration in the lower Yellow River and found that LGBM outperformed methods such as partial least squares regression, support vector regression, and random forest in terms of retrieval accuracy. Consequently, we employed the LGBM method to integrate all these variables for precipitation retrieval, effectively accounting for the spatiotemporal and physical correlations of precipitation.

In the precipitation retrieval process, we employed a two-stage strategy: precipitation event classification and precipitation value retrieval. Sixteen variables were used as independent variables in the retrieval process, and all of them are listed in **Table S3** in the supplementary materials. For the precipitation event classification model, the variable indicating whether a precipitation event occurred was used as the dependent variable, while the precipitation value was used as the dependent variable in the precipitation value retrieval model. For the convenience of updating and maintaining data every year in the

future, we constructed separate models for each year. That is, for each year, the same independent variables were used to develop two different models based on the LGBM method, with precipitation event and precipitation amount as the dependent variables, respectively. One model is used for precipitation event classification, and the other for precipitation value retrieval. From 1960 to 2023, a total of 64 years, 128 different models were generated. Specifically, for a given year, all variables required for retrieval were consolidated and split into training and validation sets at a ratio of 8:2. The training

set was utilized to develop a precipitation event classification model based on the LGBM method, while the validation set was used for hyperparameter optimization. Then, the established classification model was applied to all samples to determine whether each sample was a precipitation event. Samples identified as precipitation events were used to train a precipitation value reversal model based on the LGBM method, while non-precipitation samples were excluded from the retrieval process. This approach effectively removed the majority of non-precipitation samples, simplifying the capture of precipitation

characteristics and enhancing the accuracy of the reversal model. Additionally, this strategy notably improved the discrimination of precipitation events and mitigated the overestimation of precipitation events commonly associated with traditional interpolation-based methods. Upon completing the retrieval process, the trained precipitation value retrieval models were used to generate the final gridded daily precipitation for the entire Chinese mainland from 1960 to 2023.

**3.4 Validation**

We compared the CHM_PRE V2 precipitation dataset with five existing gridded precipitation datasets to verify its high precision and reliability. To ensure comparability, the comparison focused on the period from 2001 to 2022 for which all data have time coverage. A total of 63,397 available CMA-HD station observations were utilized to validate the accuracy of

precipitation data. There are two approaches to using station observations to validate the accuracy of gridded precipitation data. The first approach involves interpolating the station data—using methods such as IDW—to generate gridded data at the same spatial resolution as the dataset being validated. This method can produce spatially consistent results with the target gridded dataset. However, as previously mentioned, interpolation methods have some limitations and inevitably introduce interpolation-related uncertainties (McMillan et al., 2018; Wagner et al., 2012). Moreover, the uneven spatial distribution of stations makes the validation results in sparsely monitored areas less reliable. The second approach is to directly compare the station observations with the corresponding grid cell values in the dataset being validated. Although this method only provides validation results for grid cells that contain observation stations, it avoids the uncertainties introduced by interpolation and ensures the reliability of the accuracy assessment. In this study, we adopted the second approach for the validation. To align with the 0.1° gridded precipitation data, station observations were mapped onto a 0.1° grid, and the average precipitation of all stations within each grid cell was regarded as the true precipitation value for that grid cell. Metrics such as absolute error (AE), Kling-Gupta efficiency (KGE, the values range is (-∞, 1], with 1 being the optimal), and relative standard deviation (RSD, the values range is (0, +∞), with 1 being the optimal) were employed to evaluate precipitation accuracy:

$$AE = abs(y - \hat{y}) \tag{2}$$

$$KGE = 1 - \sqrt{(R(y,\hat{y}) - 1)^2 + (RSD(y,\hat{y}) - 1)^2 + (Bias(y,\hat{y}) - 1)^2} \tag{3}$$

$$RSD = \frac{\sigma_{\hat{y}}/\mu_{\hat{y}}}{\sigma_y/\mu_y} \tag{4}$$

where $y$ and $\hat{y}$ represent the observed precipitation values and the gridded precipitation values (mm/day), respectively; $\mu$ denotes the mean value, $\sigma$ signifies the standard deviation; $R$ denotes the correlation coefficient, and Bias represents the variability ratio, each defined as follows:

$$R(y,\hat{y}) = \frac{\frac{1}{N}\sum_{i=1}^{N}(y_i - \mu(y)) * (\hat{y}_i - \mu(\hat{y}))}{\sigma_y \sigma_{\hat{y}}} \tag{5}$$

$$Bias = \frac{\mu_{\hat{y}}}{\mu_y} \tag{6}$$

Precipitation errors can be categorized into systematic errors, random errors, and precipitation event detection errors (Tian et al., 2009; Wei et al., 2022). Beyond precipitation amount (systematic errors and random errors), the occurrence of precipitation events also markedly impacts hydrological modelling (Dong et al., 2020). However, commonly used precipitation accuracy metrics such as KGE and RSD only account for systematic and random errors, neglecting the precipitation event detection errors. Thus, we adopted the Heidke skill score (HSS, the values range is (0, 1], with 1 being the optimal), false alarm ratio (FAR, the values range is [0, 1], with 0 being the optimal), and Accuracy score (the values range is (0, 1], with 1 being the optimal) to assess the accuracy of precipitation event detection (AghaKouchak and Mehran, 2013; Dong et al., 2020):

$$HSS = \frac{2(TP \times TN - FP \times FN)}{(TP + FN)(FN + TN) + (TP + FP)(FP + TN)} \tag{7}$$

$$Accuracy = \frac{TP + TN}{TP + TN + FP + FN} \tag{8}$$

$$FAR = \frac{FP}{TP + FP}$$

where TP, FP, TN, and FN represent the precipitation events' matching relationship between gauged precipitation and precipitation products, with their meanings outlined in **Table S4** in the supplementary materials. The threshold for whether it is a precipitation event is more than 0.1mm of precipitation per day. Notably, to ensure the comparability of accuracy, instances where any precipitation products lack values were excluded during the accuracy calculations.

## 4 Results and discussion

### 4.1 Precipitation amount and spatial patterns

**Figure 4** illustrates the spatial distribution patterns of the multi-year average annual total precipitation for these datasets. It can be seen that all datasets exhibit similar precipitation distribution patterns, with annual totals generally decreasing from southeastern to northwestern China. Notably, CHM_PRE V2, CHM_PRE V1, GSMaP, and IMERG datasets effectively capture the high precipitation characteristics of the southern Tibetan Plateau, whereas PERSIANN-CDR and GLDAS datasets tend to underestimate precipitation in this region. Moreover, compared to satellite remote sensing-based datasets like GSMaP and IMERG, CHM_PRE V2 and CHM_PRE V1, which are based on extensive gauged observations, provide finer spatial patterns in precipitation distribution, particularly in regions with high variability, such as southeastern China. **Figure 5**(a-b) depicts the temporal characteristics of precipitation across the Chinese mainland. The various datasets show highly consistent patterns in monthly average precipitation (**Figure 5**(a)) and multi-year monthly average precipitation (**Figure 5**(b)) across all grid cells. Precipitation is higher in spring and summer (March to August), peaking in July, and lower in autumn and winter (September to February). This indicates that CHM_PRE V2 shows good consistency with the previous gridded precipitation dataset in terms of spatial patterns and temporal distribution.

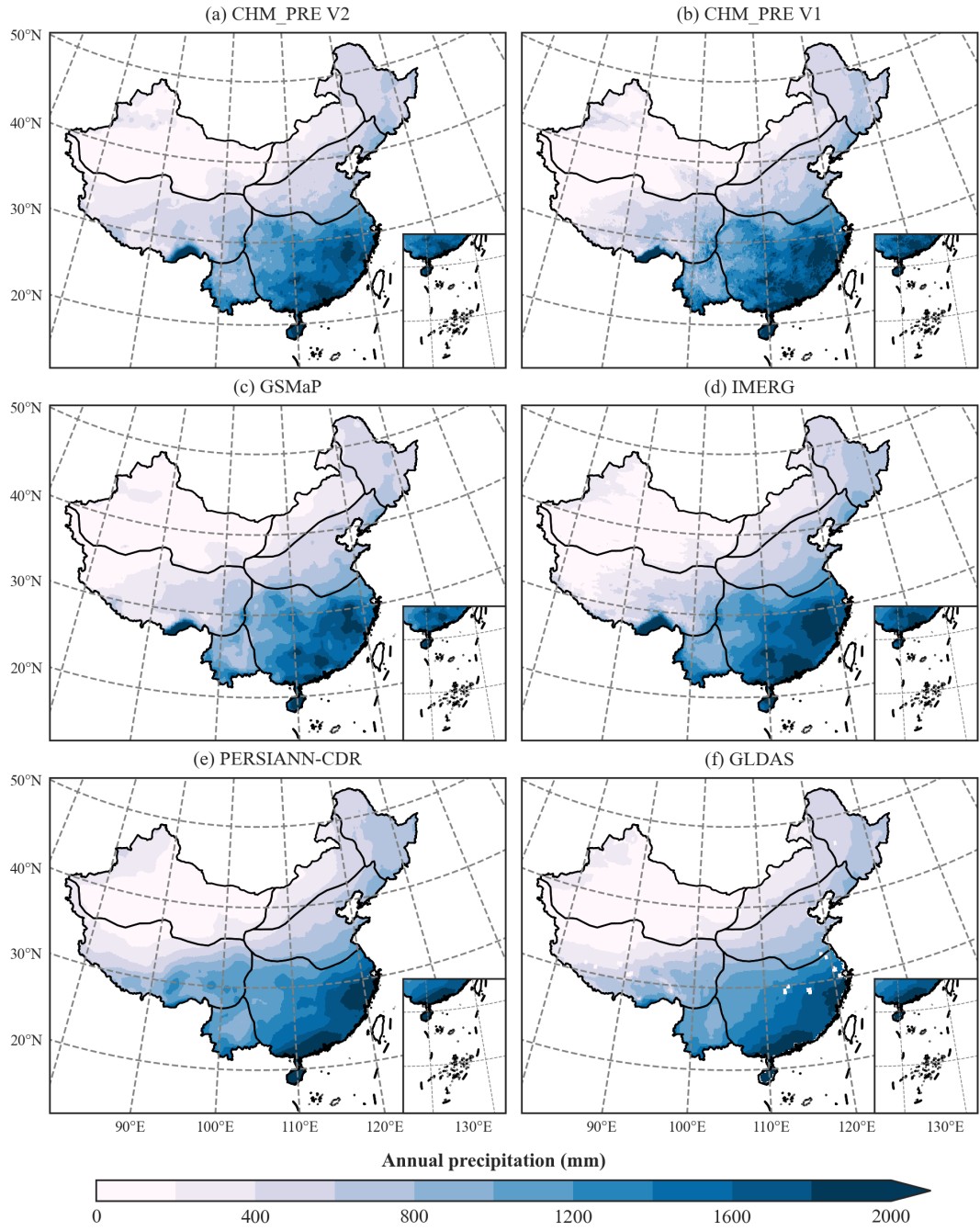

**Figure 4. Spatial distribution patterns of multi-year average annual total precipitation from 2001 to 2020.**

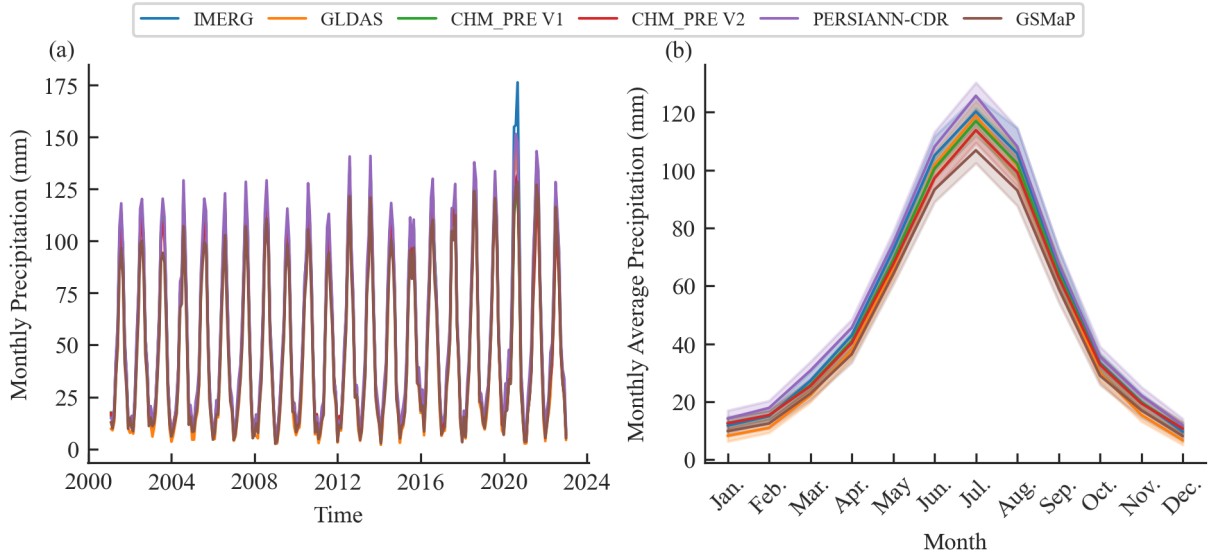

Figure 5. (a) time series of monthly precipitation; (b) multi-year mean monthly precipitation from 2001 to 2020.

## 4.2 Accuracy validation of precipitation value

**Figure 6** illustrates the overall accuracy of these datasets based on CMA-HD. Precipitation datasets derived from gauge-based interpolation (CHM_PRE V1 and CHM_PRE V2) demonstrate significantly higher accuracy compared to those based on remote sensing (GSMaP, IMERG, and PERSIANN-CDR) and reanalysis (GLDAS), as evidenced by lower absolute error, higher KGE) and RSD (**Figure 6**(a-c)). CHM_PRE V2 achieved an overall MAE, KGE, and RSD of 1.48 mm/day, 0.79, and 0.88, respectively, outperforming other datasets by 12.84%, 12.86%, and 4.76% (**Table S5** in the supplementary material).

Furthermore, the accuracy of precipitation datasets was analysed across different climatic regions. Given the superior performance of CHM_PRE V2 and CHM_PRE V1, the comparison focused exclusively on these two datasets. **Figure 6**(d-e) presents their absolute error, KGE, and RSD across different climatic regions. The results reveal a marked improvement in CHM_PRE V2's accuracy over CHM_PRE V1, with MAE increasing by 6.18% to 14.58% and KGE improving by 7.63% to 14.94% across various regions (**Figure 6**(d-e) and **Table S6** in the supplementary material). Specifically, **Figure 6**(d) shows

that both the CHM_PRE V2 and V1 datasets exhibit larger absolute errors in regions such as NC, SCC, and QT compared to other areas. This is mainly attributed to the higher precipitation amounts in these regions, which naturally lead to greater absolute errors. In contrast, accuracy metrics that are not affected by the magnitude of the variables, such as KGE (**Figure 6**(e)) and RSD (**Figure 6**(f)), demonstrate better stability across different regions. The KGE and RSD in SWC and QT exhibit relatively greater variability, which could possibly be explained by the sparse distribution of precipitation observation

stations and the high spatiotemporal variability of precipitation in these regions (Li et al., 2015; Liu et al., 2019).

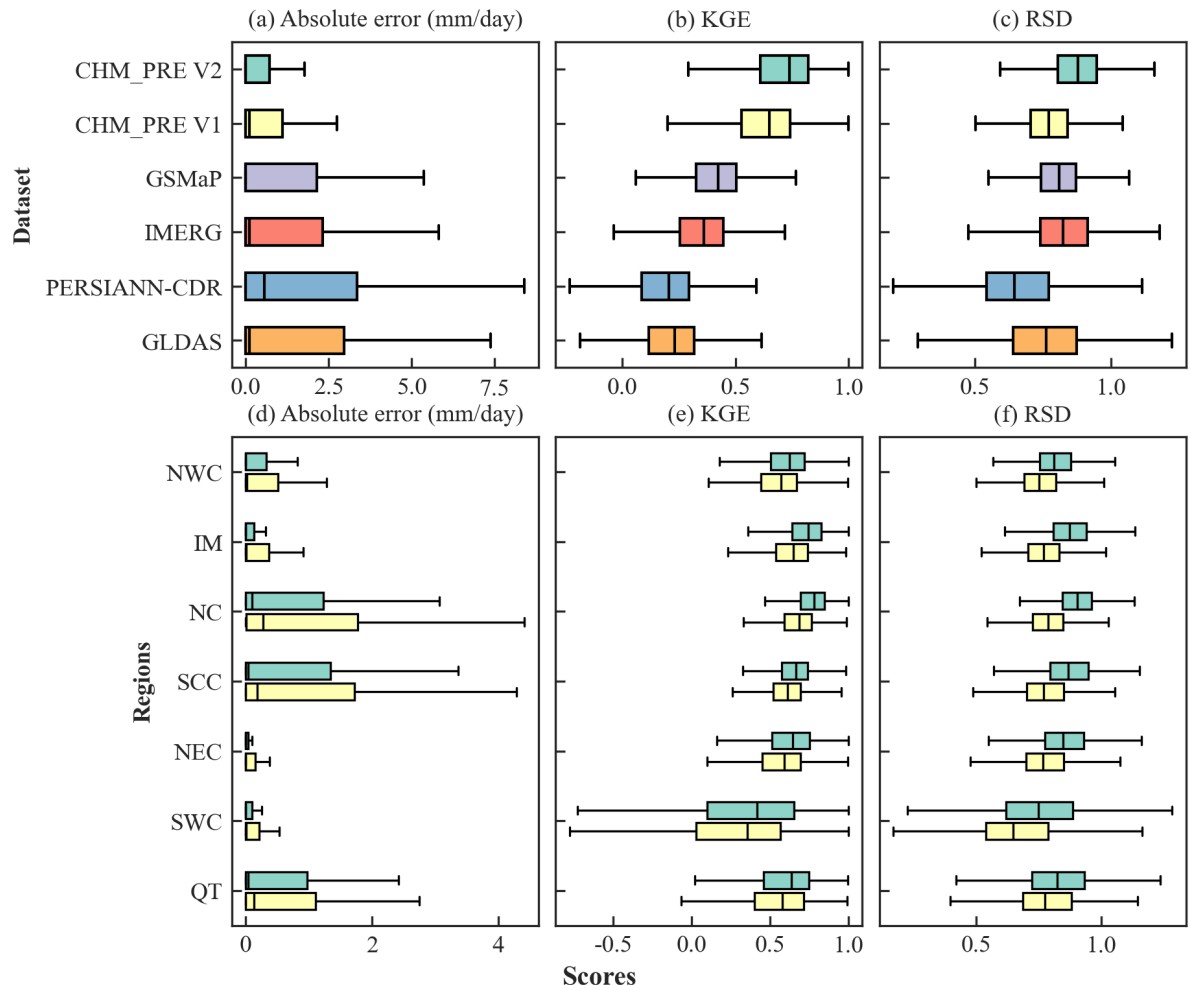

**Figure 6. Accuracy of different precipitation datasets on the testing dataset CMA-HD. The green and yellow boxes in subfigures (d-f) represent CHM_PRE V2 and CHM_PRE V1, respectively. The ideal values for absolute error, KGE, and RSD are 0 mm/day, 1.0, and 1.0, respectively.**


Further comparison at the grid scale of the three precipitation datasets with the relatively highest accuracy (CHM_PRE V2, CHM_PRE V1, and GSMaP) was conducted. **Figure 7** illustrates the spatial distribution of KGE and RSD for the three datasets. CHM_PRE V2 demonstrates a significant improvement in KGE compared to CHM_PRE V1 and GSMaP, with many grid cells in the NWC and IM regions showing an increase from below 0.2 to above 0.4, and numerous grid cells in the

SCC and NC regions rising from the 0.6–0.8 range to above 0.8. The median KGE value of CHM_PRE V2 across all grid cells reaches 0.738, representing an approximate 13.87% improvement over CHM_PRE V1. Regarding RSD, GSMaP's accuracy slightly outperforms CHM_PRE V1; however, CHM_PRE V2 exhibits a distinct advantage over the other datasets, with a median RSD value of 0.880, reflecting an 8.64% enhancement compared to the other datasets.

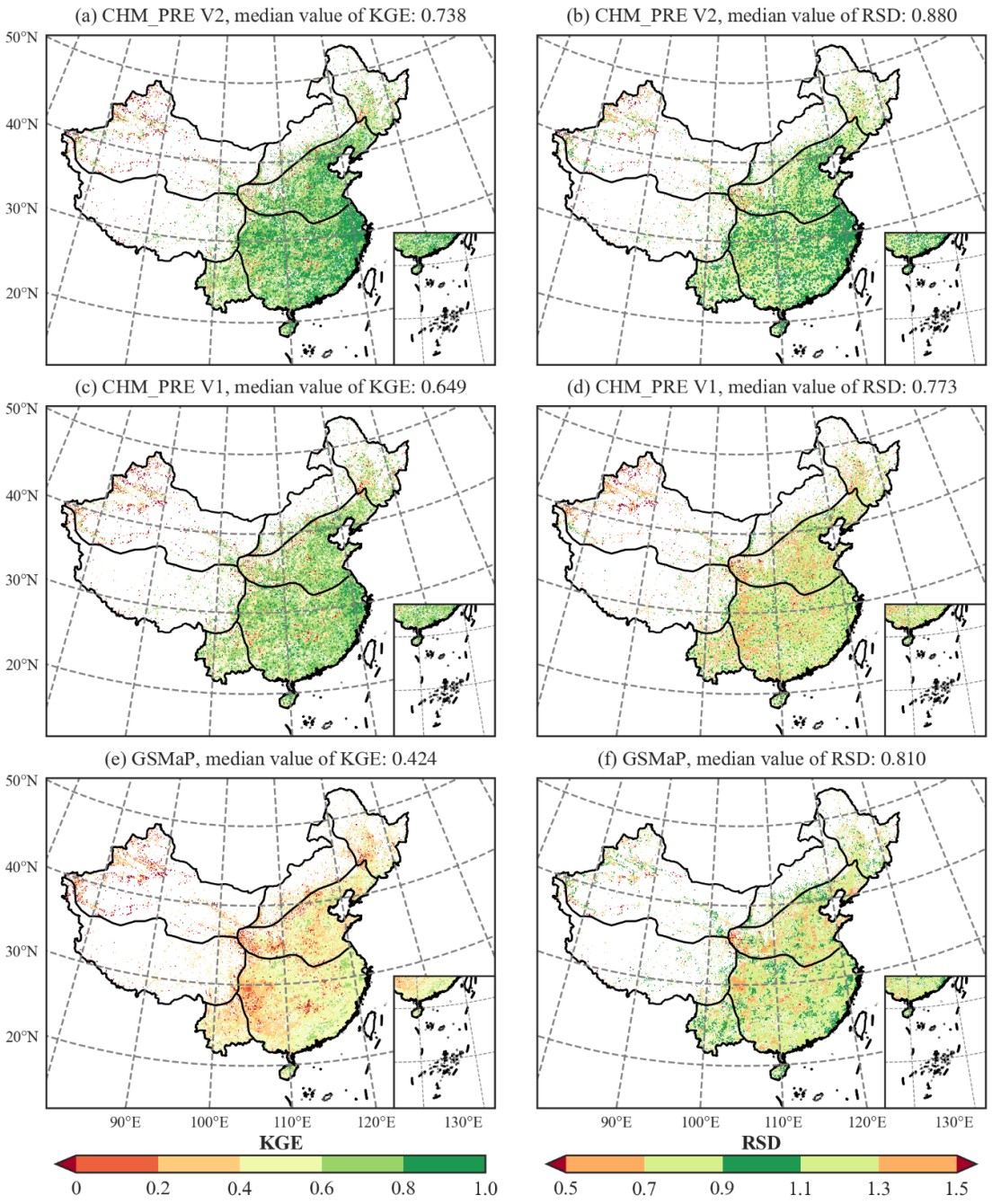

**Figure 7. Accuracy of different precipitation datasets at each grid cell on the testing data CMA-HD. (a), (c) and (e) show the KGE of each grid for CHM_PRE V2, PRE V1, and GSMaP, respectively; (b), (d) and (f) show the RSD of each grid for CHM_PRE V2, PRE V1, and GSMaP, respectively.**

### 4.3 Accuracy validation of precipitation event detection capability

**Figure 8**(a-c) illustrates the HSS, Accuracy score, and FAR metrics, evaluated using CMA-HD across different datasets.
CHM_PRE V2 demonstrates a significantly superior ability to capture precipitation events across all three metrics compared to other precipitation datasets. Specifically, CHM_PRE V2 achieves an overall HSS of 0.68, an Accuracy score of 0.85, and a FAR of 0.24, surpassing other datasets by approximately 17.24%, 7.59%, and 29.17%, respectively (**Table S7** in the supplementary materials). Notably, a lower FAR value indicates better performance, with 0 being optimal, which distinguishes it from the other two metrics. Similarly, we analysed the precision of CHM_PRE V2 and CHM_PRE V1 in
capturing precipitation events across different climatic regions. **Figure 8** (d-f) and **Table S8** in the supplementary materials reveal that CHM_PRE V2 consistently outperforms CHM_PRE V1 across all regions. The overall HSS values for CHM_PRE V2 in different regions reach 0.52–0.68, representing an improvement of approximately 10.16% to 22.98% over CHM_PRE V1. Further analysis of the FAR and probability of detection (POD) metrics shows that CHM_PRE V2 achieves improvements in FAR by 15.73% to 70.79% compared to CHM_PRE V1 across different climatic regions. However, the
POD values for CHM_PRE V2 decrease by approximately 6.79% to 11.25% compared to CHM_PRE V1. This indicates that the improved accuracy of CHM_PRE V2 in capturing precipitation events is primarily due to a reduction in overestimation, attributable to the two-stage retrieval approach described in Section 3.3.

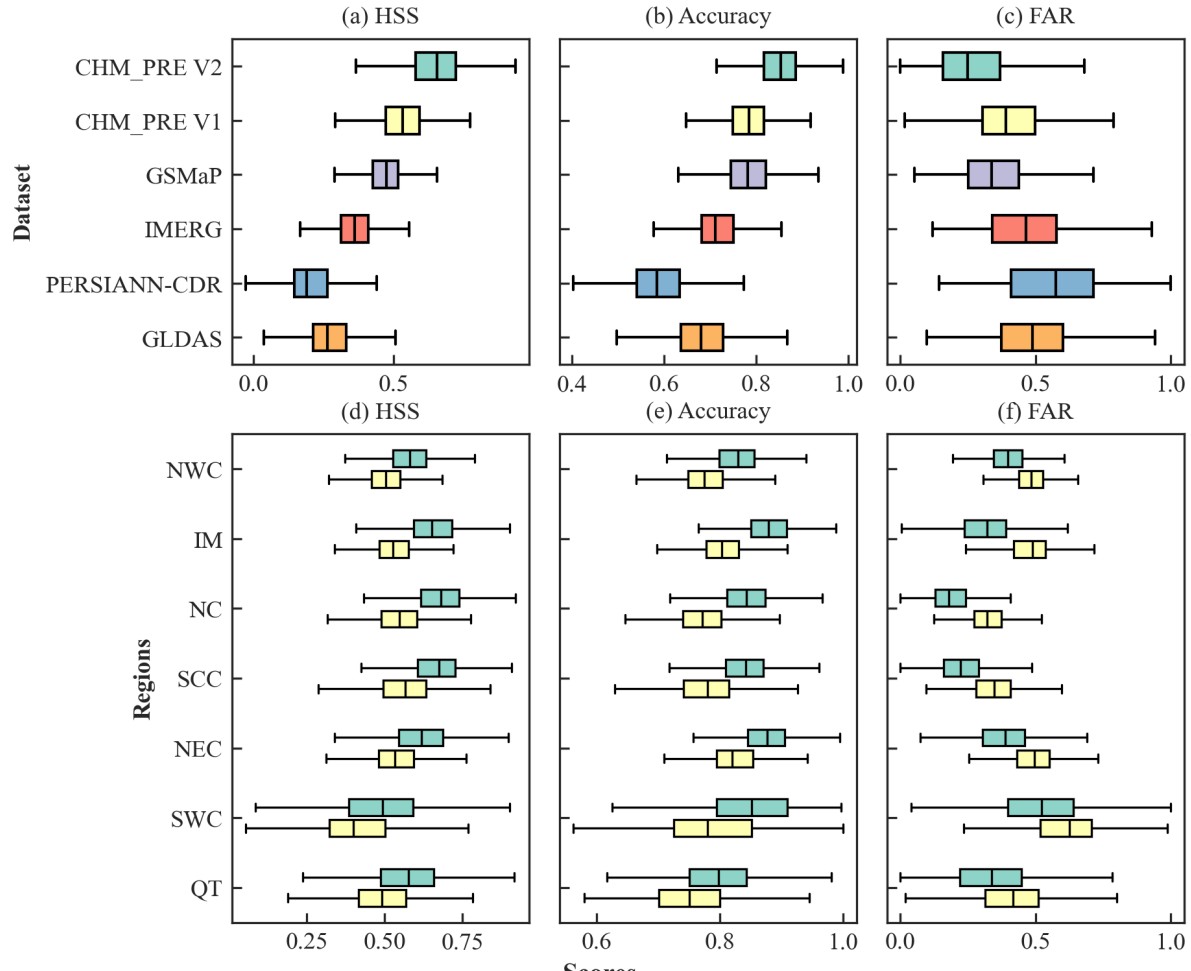

**Figure 8. Accuracy precipitation events for different precipitation datasets on the testing dataset CMA-HD. The green and yellow boxes in subfigures (d-f) represent CHM_PRE V2 and CHM_PRE V1, respectively.**

We further analyse the accuracy of precipitation events from the CHM_PRE V2, CHM_PRE V1, and GSMaP datasets across different grids. **Figure 9** illustrates the spatial distribution of the HSS and Accuracy scores for the three datasets. The KGE for CHM_PRE V2 shows a significant improvement over both CHM_PRE V1 and GSMaP, with the HSS values for many grid cells rising from 0.2–0.6 to 0.6–0.8. The total HSS across all grid cells reaches 0.654, representing a 23.16% improvement compared to other datasets. Regarding the Accuracy score, it is evident that GSMaP outperforms CHM_PRE V1 in regions such as NWC, NEC, and IM, while CHM_PRE V1 surpasses GSMaP in regions like SCC and NC. In contrast, CHM_PRE V2, which combines the advantages of interpolation-based and remote sensing-based precipitation data, outperforms all other datasets across all regions.

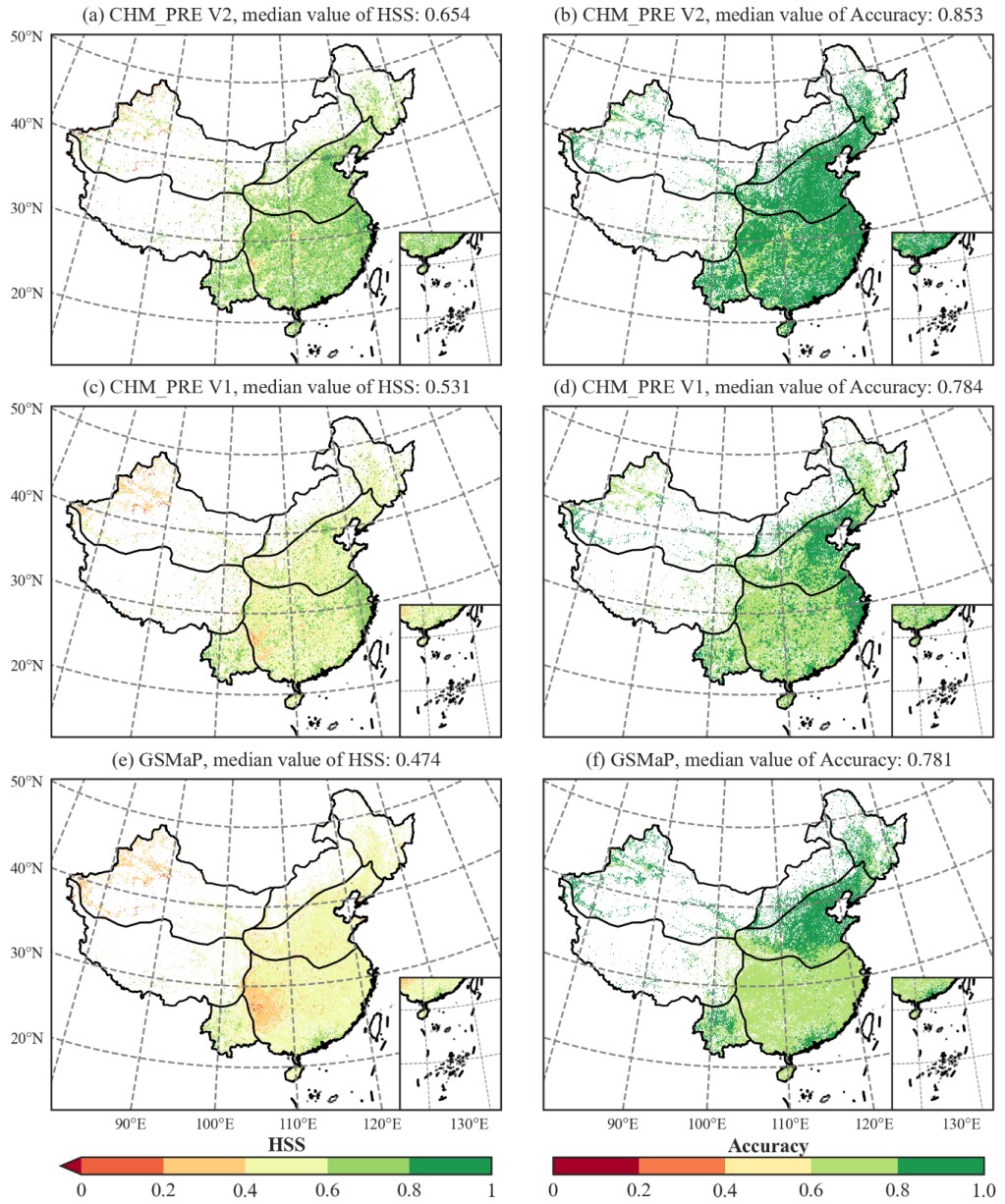

**Figure 9.** Accuracy of precipitation events for different precipitation datasets at each grid cell on the testing data CMA-HD. (a), (c) and (e) show the HSS of each grid for CHM_PRE V2, PRE V1, and GSMaP, respectively; (b), (d) and (f) show the Accuracy score of each grid for CHM_PRE V2, PRE V1, and GSMaP, respectively.

## 4.4 Improvements compared to the previous CHM_PRE V1 dataset

390 CHM_PRE V2 is a continuation and improvement of our previously published CHM_PRE V1. Therefore, we further summarize the differences between CHM_PRE V2 and CHM_PRE V1 in **Table 1**, and highlight the improvements of

CHM_PRE V2 over the previous version by using bold font. It can be observed that CHM_PRE V2 shares the same spatiotemporal resolution and coverage with V1 (except for the extended time range up to 2023), mainly to maintain consistency with other datasets in the CHM family (Zhang et al., 2025). The spatial interpolation method used in CHM_PRE V2 is largely consistent with that in V1, but it incorporates precipitation-related covariates in a data-driven manner by integrating the LGBM method. Eleven precipitation-related variables were considered, including topographic features (elevation and slope), satellite-derived precipitation estimates, reanalysis-based precipitation products, soil moisture, NDVI, recent daily precipitation records, and aggregate precipitation metrics. The inclusion of these covariates allows for a better representation of the spatiotemporal variability of precipitation (Gu et al., 2023; Ma et al., 2025), resulting in improved precipitation accuracy (with MAE and KGE reaching 1.48 mm/day and 0.79, representing improvements of approximately 12.84% and 12.86% compared to CHM_PRE V1, respectively). In addition, the capability of detecting precipitation events is a critical indicator of the accuracy of precipitation datasets (Dong et al., 2020; Kang et al., 2024). CHM_PRE V2 applies a two-stage modelling approach to distinguish and correct precipitation events, which reduces overestimation of such precipitation events and improves event detection accuracy (with FAR and HSS reaching 0.24 and 0.68, respectively, reflecting improvements of approximately 54.17% and 17.24% over CHM_PRE V1). Overall, CHM_PRE V2 demonstrates obvious improvements over CHM_PRE V1 and serves as a high-accuracy daily gridded precipitation dataset for the Chinese mainland.

**Table 1. Comparison between CHM_PRE V2 and CHM_PRE V1.**

| Category | Item | CHM_PRE V1 | CHM_PRE V2 |
|---|---|---|---|
| Metadata | Spatial resolution | 0.1° | 0.1° |
| | Temporal resolution | Daily | Daily |
| | Spatial coverage | 18°N–54°N, 72°E–136°E | 18°N–54°N, 72°E–136°E |
| | Time Span | 1961–2022 | 1960–2023 |
| Method | Spatial autocorrelation considered | ✓ | ✓ |
| | Interpolation method | Improved IDW method | Improved IDW method |
| | Precipitation-related covariates | Only PRISM climatology data | **11 precipitation covariates** |
| | Covariate modelling approach | ✗ | **LGBM** |
| | Precipitation event considered | ✗ | ✓ |
| Accuracy of precipitation value | MAE (mm/day) | 1.67 | **1.48** |
| | KGE | 0.70 | **0.79** |
| | RSD | 0.78 | **0.88** |
| Accuracy of precipitation event | HSS | 0.58 | **0.68** |
| | Accuracy score | 0.79 | **0.85** |
| | FAR | 0.37 | **0.24** |

## 5 Data availability

The CHM_PRE V2 dataset provides daily precipitation data with a resolution of 0.1°, covering the entire Chinese mainland (18°N–54°N, 72°E–136°E). This dataset covers the period of 1960–2023, and will be continuously updated annually. The daily precipitation data is provided in NetCDF format, and for the convenience of users, we also offer annual and monthly total precipitation data in both NetCDF and GeoTIFF formats. All of these data can be freely accessed at https://doi.org/10.5281/zenodo.14632156 (Hu and Miao, 2025).

## 6 Conclusions

In this study, we developed a new high-resolution daily gridded precipitation dataset for the Chinese mainland (CHM_PRE V2) covering the period from 1960 to 2023 at a 0.1° spatial resolution. This dataset was constructed using long-term precipitation observations from 3,746 gauges and 11 carefully selected precipitation covariates. By integrating an improved inverse distance weighting interpolation method with a machine learning-based light gradient boosting machine (LGBM) algorithm, our approach accounts for spatial autocorrelation and a broad suite of covariates that represent both environmental and climatic influences on precipitation variability. The resulting CHM_PRE V2 dataset was compared with five existing gridded precipitation datasets and validated for accuracy using precipitation data from over 63k automated rain gauge stations. The results demonstrate that CHM_PRE V2 aligns closely with the overall spatiotemporal distribution patterns of existing gridded precipitation datasets, while achieving substantial improvements in precipitation event detection and precipitation value estimation. Specifically, compared to the previous dataset with the highest accuracy (CHM_PRE V1), CHM_PRE V2 achieves a 12.84% reduction in mean absolute error and a 12.86% improvement in Kling-Gupta efficiency, reaching 1.48 mm/day and 0.88, respectively. In terms of precipitation event capture, CHM_PRE V2 achieves an overall Heidke skill score, Accuracy Score, and false alarm ratio of 0.68, 0.85, and 0.24, respectively—improving upon reference datasets by 17.24%, 7.59%, and 29.17%, respectively. Particularly in the precipitation-heavy regions of north China and central-south China, the false alarm ratio reduction reaches 53.33% and 68.42%, significantly reducing the overestimation of precipitation events. These findings prove that CHM_PRE V2 is a high-precision precipitation dataset, offering substantial support for various studies in hydrology, climatology, and climate change research.

## Author contributions

JH and CM contributed to designing the research; JH implemented the research and wrote original draft; CM supervised the research; all co-authors revised the manuscript and contributed to the writing.

**Competing interests**

The contact author has declared that none of the authors has any competing interests.

**Disclaimer**

Publisher's note: Copernicus Publications remains neutral with regard to jurisdictional claims made in the text, published maps, institutional affiliations, or any other geographical representation in this paper. While Copernicus Publications makes every effort to include appropriate place names, the final responsibility lies with the authors. Regarding the maps used in this paper, please note that Figures 2, 4, 7, and 9 contain disputed territories.

**Acknowledgments**

This research was supported by the National Natural Science Foundation of China (No. U24A20572), the National Key Research and Development Program of China (No. 2024YFF0809301) and the Fundamental Research Funds for the Central Universities. We would like to express our gratitude to the China Meteorological Administration (CMA, http://data.cma.cn) for providing the observed climate data of China. We also extend our thanks to NOAA, NASA, CGIAR-CSI, and JAXA for their valuable contribution of publicly available research data that supported the completion of this work.

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
