# Peer review of "An upgraded high-precision gridded precipitation dataset for the Chinese mainland considering spatial autocorrelation and covariates"

_Earth System Science Data, 2025_

## Author Response (AR1)

**A DETAILED LIST OF THE RESPONSES**
**TO REVIEWER #1**

**Anonymous Reviewer #1 comments**

**General comments:**

This study introduces a new precipitation dataset, CHM_PRE V2, for China, demonstrating notable accuracy improvements over its predecessor, CHM_PRE V1, as well as several other existing precipitation datasets. The work represents a valuable contribution to the field, particularly for researchers seeking high-quality precipitation data in China, and is well-suited for publication in ESSD. My comments are as below. Hope those can help the authors further improve the manuscript.

**Response:** Thank you very much for the positive comments. We have made substantial revisions according to your suggestions, which have been very valuable in improving the manuscript. We hope these changes meet your expectations.

**Specific comments:**

1. My only concern is about a terminology used in this manuscript. The phrase "interpolation considering spatiotemporal and physical correlations" appears to introduce a new term for a method that has been widely used in prior research. While the authors aim to highlight the integration of spatial, temporal, and physical factors in their interpolation approach, the study does not explicitly quantify real correlation coefficients or provide a transparent framework for how these correlations are incorporated; instead, it employs a black-box approach where the spatiotemporal and physical correlations are not directly tangible.

For precipitation estimation, precipitation can be treated as a predictand, with various predictors such as static variables (latitude, longitude, elevation, slope) and dynamic variables (gridded precipitation datasets, soil moisture, precipitation climatology) as outlined in Table 1. Categorizing them strictly into spatial, temporal, and physical correlations (as done in Table 1 and elsewhere in the manuscript) may not accurately reflect their complex interdependencies. Many variables exhibit overlapping spatial, temporal, and physical correlations simultaneously. In addition, classifying GLDAS and satellite precipitation under "physical correlation" does not make sense, as it essentially implies that "precipitation correlates with precipitation". Your approach looks more like merging multiple sources of precipitation data.

Additionally, the physical correlation of NDVI on a daily scale is questionable and warrants further justification. Vegetation does not show immediate response to precipitation. Furthermore, the importance analysis based on these correlation classifications may not be reliable. For instance, if additional features are added to a specific category, I think this could artificially inflate the perceived importance of that category (e.g., Figure 5d).

Given these concerns, I recommend that the authors avoid introducing a new term that may not accurately describe the method's nature, especially given the extensive body of research on precipitation estimation. The authors approach of using new predictors (i.e., Table 1) can benefit accuracy improvement, while this falls within the feature engineering field which can be clarified in the manuscript.

**Response:** Thank you for your insightful comments. In the latest manuscript, we have removed the previous, imprecise description of "spatiotemporal and physical correlations." The term "spatial correlation" has been updated to "spatial autocorrelation" to more accurately express the dependence of precipitation at a location on surrounding areas. Additionally, "temporal and physical correlations" have been revised to "precipitation-related covariates." We have modified all relevant parts of the manuscript accordingly.

Regarding the relative importance of covariates to precipitation retrieval (Figure 5(c)), as you rightly pointed out, the importance analysis results may not be sufficiently reliable. To maintain the rigor of the manuscript, we have removed this part from the revised version. The major revisions are as follows:

[revised manuscript text omitted]

Thank you again for your thoughtful comments and support, which have helped us significantly improve the rigor of our manuscript.

2. About the title, the two words "new upgraded" seems repeated.

**Response:** According to your suggestion, we have revised the title to "An upgraded high-precision gridded precipitation dataset for the Chinese mainland considering spatial autocorrelation and covariates."

3. There are three versions of datasets published at https://zenodo.org/records/14634575. What's the difference among them?

**Response:** We appreciate your careful comment. In fact, there are no substantive differences among the multiple versions of the dataset on Zenodo. Zenodo requires the creation of a new version whenever any file within a dataset is modified. During the data upload process, we

updated the content of the documentation file (in PDF format), which necessitated the creation of multiple dataset versions. To help users better understand the differences among these versions, we have added corresponding explanations on Zenodo. Furthermore, we have updated the dataset link provided in the manuscript (https://doi.org/10.5281/zenodo.14632156) to one that will always resolve to the latest version of the dataset.

4. Line 23 and 25: I suppose those improvements use CHM_PRE V1 as a benchmark. Please specify this.
**Response**: We apologize for not making this point clear in the previous manuscript. In this study, we compared CHM_PRE V2 with five existing gridded precipitation datasets and calculated the improvement ratio of CHM_PRE V2 relative to the best-performing dataset among them. Our previous CHM_PRE V1 dataset did not always outperform the other datasets across all evaluation metrics. For example, in terms of the false alarm ratio (FAR) metric, GSMaP performed slightly better than CHM_PRE V1 (**Table S7**). Therefore, the comparison was not always based on CHM_PRE V1 as the benchmark. To clarify this point, we have revised the corresponding description as follows:

"Specifically, it achieves a mean absolute error of 1.48 mm/day and a Kling-Gupta efficiency of 0.88, representing improvements of 12.84% and 12.86%, respectively, compared to the previously optimal dataset. Regarding precipitation event detection, CHM_PRE V2 achieved a Heidke skill score of 0.68 and a false alarm ratio of 0.24, surpassing the previously optimal dataset by 17.24% and 29.17%, respectively." (Lines 23–27)

**Table S7**. Precipitation event accuracy of different datasets validated by high-density gauge data. The bolded numbers in the column represent the optimal accuracy values for that metric.

| Dataset Name | HSS | F1 Score | Accuracy | POD | FAR |
|---|---|---|---|---|---|
| CHM_PRE V2 | **0.68** | **0.80** | **0.85** | 0.84 | **0.24** |
| CHM_PRE V1 | 0.58 | 0.75 | 0.79 | **0.93** | 0.37 |
| GSMaP | 0.50 | 0.67 | 0.78 | 0.65 | 0.31 |
| IMERG | 0.39 | 0.62 | 0.71 | 0.69 | 0.43 |
| PERSIANN-CDR | 0.21 | 0.54 | 0.59 | 0.70 | 0.55 |
| GLDAS | 0.29 | 0.54 | 0.68 | 0.55 | 0.47 |

5. Please introduce the methodological difference between V2 and V1 datasets in the abstract.
**Response**: According to your suggestion, we have rewritten the abstract to highlight the methodological difference of CHM-PRE V2 compared to V1, as follows:

"Building upon the improved inverse distance weighting interpolation method used in our previous dataset CHM_PRE V1, we integrated a machine learning algorithm—light gradient boosting machine (LGBM)—to incorporate precipitation-related covariates in a data-driven manner. This integration allows for a more comprehensive characterization of precipitation

patterns, jointly capturing spatial autocorrelation and covariate-based variability." (Lines 17–21)

6. Line 26: What do the three numbers represent?

**Response:** We apologize for not clearly conveying this point in the previous manuscript. The original sentence — "Feature importance analysis revealed that spatiotemporal and physical correlations contributed 37.10%, 34.11%, and 28.78% to precipitation retrieval, underscoring the necessity of incorporating temporal and physical correlations." — referred to the relative contributions of spatial, temporal, and physical correlations to precipitation retrieval (previously shown in Figure 5(c)). However, as you pointed out in Comment 1, the results of this importance analysis may not be sufficiently reliable. To ensure the rigor of the manuscript, we have removed this part in the latest manuscript (latest Figure 5).

Thanks again for your valuable comments.

[Figure]

**Latest Figure 5**. (a) time series of monthly precipitation; (b) multi-year mean monthly precipitation from 2001 to 2020.

[Figure]

**Previous Figure 5**. (a) time series of monthly precipitation; (b) multi-year mean monthly precipitation from 2001 to 2020; (c) feature importance of precipitation retrieval. In the figure, precipitation is abbreviated as "Prec.," interpolation-based precipitation is denoted as "Interp. Prec.," while remote sensing and soil moisture are represented by "RS" and "SM," respectively; "1st-day prior Prec." to "5th-day prior Prec." means the precipitation from the 1st day ago to 5th day ago.

7. Line 51: What interpolation method?

**Response**: In response to your suggestion, we have made the following revisions to improve the clarity of the manuscript:

"Our previous study developed a gridded precipitation dataset for the Chinese mainland (a member of the China Hydro-Meteorology datasets, hereinafter called CHM_PRE V1) based on inverse-distance weighting interpolation method and parameter-elevation regression on independent slopes model (PRISM) (Daly et al., 1994, 2002), using data from 2,839 gauges. The CHM_PRE V1 demonstrates overall high accuracy across the Chinese mainland (Han et al., 2023), and has received widespread attention and extensive use, benefiting a large number of hydro-meteorological related studies (Hu et al., 2024; Wan and Zhou, 2024; Yin et al., 2025)." (Lines 50–55)

8. Line 60: "are" should be "is". Besides, putting "historical precipitation data" in this sentence is weird.
**Response**: We have thoroughly rewritten these sentences and the results are as follows

"In summary, a key limitation of existing datasets is that they tend to focus on either spatial autocorrelation or a limited set of precipitation-related covariates, but rarely incorporate multiple types of information simultaneously. However, precipitation is influenced not only by spatial autocorrelation—that is, the dependence of precipitation at a given location on surrounding areas (Chen et al., 2010, 2016; Fan et al., 2021; Huff and Shipp, 1969; Tang et al.,

2020)—but also by a wide array of covariates, such as elevation, land surface conditions, atmospheric parameters, and recent precipitation events (Adler et al., 2008; Ham et al., 2023; Ravuri et al., 2021; Trucco et al., 2023)." (Lines 57–62)

9. Line 95: I think the station data cannot be freely accessed from this website ...
**Response:** Thank you very much for this valuable feedback. We obtained the precipitation gauge data from the China Meteorological Administration in January 2024. To better clarify this point, we have corrected the corresponding description in the manuscript:

"The primary daily precipitation gauge data sourced from the China Meteorological Administration (CMA; http://data.cma.cn, last access: January 2024) spans the entire Chinese mainland, encompassing records from 2,816 stations between 1960 and 2023." (Lines 95–97)

10. Line 222-223: This does not seem to be solid reason for selecting LGBM.
**Response**: Thank you for your helpful comment. Our previous research (Hu et al., 2023) has demonstrated that the LGBM method achieves a higher accuracy compared to other commonly used machine learning methods (such as Random Forest and Support Vector Machine). Therefore, we adopted the LGBM method as the retrieval method in this study. Following your suggestion, we have provided a more detailed explanation of the reason for choosing the LGBM method. The corresponding revisions are as follows:

"LGBM demonstrates exceptional accuracy and generalization, making it widely applicable to various tasks such as classification, regression, and ranking (Bian et al., 2023; Jiang et al., 2024; Zhang et al., 2024). Hu et al. (2023) applied LGBM to the retrieval of suspended sediment concentration in the lower Yellow River and found that LGBM outperformed methods such as partial least squares regression, support vector regression, and random forest in terms of retrieval accuracy. Consequently, we employed the LGBM method to integrate all these variables for precipitation retrieval, effectively accounting for the spatiotemporal and physical correlations of precipitation." (Lines 238–243)

11. Section 3.3: The description of data training is unclear to me. I recommend that the users use a few bullet points to explain what are the inputs and outputs of CHM_PRE production. For example, after reading Section 3.3 and looking at Table 1, I am still not sure what are samples you used in model training.
**Response**: We apologize for not clearly describing the modeling process in the previous manuscript. In the latest manuscript, we have thoroughly rewritten Section 3.3 and added **Table S3** in the supplementary materials to better illustrate the modeling process and the modeling variables. We hope that the revised Section 3.3 meets your expectations. The corresponding revisions are as follows:

[revised manuscript text omitted]

Thank you again for your valuable comments, which have greatly helped us improve the quality of the manuscript.

[revised manuscript text omitted]

------------------------------------------------ end line-------------------------------------------------------

In order to make the review of our revision more convenient, we have marked all changes using the "Track Changes" function in Microsoft Word, and have uploaded the "tracked changes" version as Supplementary Material.

**A DETAILED LIST OF THE RESPONSES**
**TO REVIEWER #2**

**Anonymous Reviewer #2 comments**
**General comments:**

The study presents a novel and interesting approach to developing high-resolution gridded precipitation data (CHM_PRE v2.0) by integrating station data, multiple covariate factors, and machine learning techniques. Given the significant spatial and temporal variability of precipitation, the development of reliable and credible gridded precipitation datasets is crucial for hydroclimatological research. The authors have clearly put considerable effort into this work, particularly by incorporating covariate factors beyond traditional spatial interpolation methods. This dataset is likely to have broad applicability in the field. However, several issues need to be addressed to improve the clarity, rigor, and impact of the manuscript.

**Response**: We greatly appreciate your careful reading of the manuscript, insightful comments, and valuable suggestions. Your thoughtful review has enhanced our paper considerably. The manuscript has been revised thoroughly according to your comments, with our point-by-point responses detailed below.

**Specific comments:**

1. The introduction mentions that the authors previously developed CHM_PRE v1.0. It is unclear how much innovation or improvement has been achieved in v2.0 compared to v1.0. The authors should provide a detailed explanation of the differences between the two versions and justify why a new release (v2.0) is necessary instead of simply updating v1.0. This is critical for readers to understand the added value of this new version.

**Response**: Thank you for the valuable comment. We have revised the introduction to better highlight the differences between CHM_PRE V2 and V1, as well as the significance of CHM_PRE V2. The corresponding revisions are as follows:

"Our previous study developed a gridded precipitation dataset for the Chinese mainland (a member of the China Hydro-Meteorology datasets, hereinafter called CHM_PRE V1) based on inverse-distance weighting interpolation method and parameter-elevation regression on independent slopes model (PRISM) (Daly et al., 1994, 2002), using data from 2,839 gauges. The CHM_PRE V1 demonstrates overall high accuracy across the Chinese mainland (Han et al., 2023), and has received widespread attention and extensive use, benefiting a large number of hydro-meteorological related studies (Hu et al., 2024; Wan and Zhou, 2024; Yin et al., 2025). However, interpolation-based precipitation datasets rely heavily on ground meteorological gauges, performing poorly in areas with sparse station distribution or missing data.

In summary, a key limitation of existing datasets is that they tend to focus on either spatial autocorrelation or a limited set of precipitation-related covariates, but rarely incorporate multiple types of information simultaneously. However, precipitation is influenced not only by spatial autocorrelation—that is, the dependence of precipitation at a given location on surrounding areas (Chen et al., 2010, 2016; Fan et al., 2021; Huff and Shipp, 1969; Tang et al.,

2020)—but also by a wide array of covariates, such as elevation, land surface conditions, atmospheric parameters, and recent precipitation events (Adler et al., 2008; Ham et al., 2023; Ravuri et al., 2021; Trucco et al., 2023). This lack of comprehensive consideration for multiple covariates constrains the accuracy of these datasets, particularly in regions with sparse meteorological stations, such as western China (Jiang et al., 2023). Moreover, existing methods tend to generate excessive minor precipitation, leading to an overestimation of precipitation events, which will have considerable impacts on hydrologic modelling (Dong et al., 2020; Kang et al., 2024; Wei et al., 2022).

To address the aforementioned issues, this study introduces a new high-precision, long-term daily gridded precipitation dataset for the Chinese mainland (a member of the China Hydro-Meteorology datasets, hereinafter called CHM_PRE_V2). Building on CHM_PRE_V1, CHM_PRE V2 integrates precipitation gauges, remote sensing observations, reanalysis data, and various precipitation-related factors. Through the use of advanced spatial interpolation and machine learning algorithms, our method captures spatial autocorrelation while jointly modelling multiple covariates to enhance precipitation accuracy." (Lines 50–72)

2. The authors fused data from 2,816 stations to produce 0.1-degree gridded precipitation data. However, the rationale for choosing 0.1-degree resolution over finer resolutions (e.g., 0.05-degree or 1 km) is not explained. Given the availability of high-resolution precipitation datasets in China, including sub-daily data, the authors should discuss why 0.1-degree resolution was selected and whether finer resolutions were considered.

**Response**: Thank you for pointing out the issue regarding the choice of resolution. Spatial resolution is a very important attribute for gridded datasets, and in this study, consistency with our previous datasets was the primary factor in selecting the spatial resolution. Our previous datasets, CHM_Drought (Zhang et al., 2025) and CHM_PRE V1 (Han et al., 2023), both have a spatial resolution of 0.1°. Therefore, CHM_PRE V2 was also set at this resolution to ensure compatibility with other datasets in the CHM family. In addition, we believe that a 0.1° resolution provides a good balance between accuracy and computational efficiency at large scales. We have added some explanations in the introduction to better clarify this point:

"The spatial resolution of the dataset is set to 0.1° to maintain consistency with our previous dataset (Han et al., 2023; Zhang et al., 2025)." (Lines 73–74)

3. The terms "Spatiotemporal correlated data" and "Physically correlated data" are introduced but not clearly defined. A more detailed explanation of these terms is necessary to ensure readers fully understand the methodology and its theoretical basis.

**Response**: We sincerely appreciate your insightful comments. As you pointed out, the explanation of "spatiotemporal and physical correlations" in the previous manuscript was unclear. Upon careful consideration, we concluded that introducing these definitions is not necessary. Therefore, in the latest manuscript, we have updated "spatial correlation" to "spatial autocorrelation" to more precisely express the dependence of precipitation at a location on surrounding areas (Chen et al., 2010, 2016; Fan et al., 2021; Huff and Shipp, 1969). Meanwhile, "temporal and physical correlations" have been revised to "precipitation-related covariates."

We have made corresponding revisions throughout the manuscript wherever correlations were mentioned to ensure the rigor of the manuscript. The major revisions are summarized as follows:

[revised manuscript text omitted]

Thank you again for your thoughtful comments and support, which have helped us significantly improve the rigor of our manuscript.

4. The manuscript contains numerous abbreviations, which hinder the readability and flow of the text. The authors should minimize the use of abbreviations or provide a glossary for reference.
**Response**: Thank you for pointing out the issue regarding the abbreviations. Following your suggestion, we have summarized all abbreviations used in the manuscript and added them to the supplementary materials (**Table S1**). The corresponding revisions are as follows:

"For clarity, a list of abbreviations used throughout this paper is presented in **Table S1** in the supplementary materials." (Lines 77–78)

**Table S1**. List of abbreviations used throughout this paper.

| Abbreviation | Full Term |
|---|---|
| LGBM | Light gradient boosting machine |
| PRISM | Parameter-elevation regression on independent slopes model |
| IDW | Inverse distance weighting |
| CMA | China Meteorological Administration |
| GHCND | Global Historical Climatology Network-Daily |
| NCDC | National Climatic Data Center |
| NOAA | National Oceanic and Atmospheric Administration |
| SRTM | Shuttle Radar Topography Mission |
| DEM | Digital Elevation Model |
| CGIAR-CSI | Consortium for Spatial Information, Consultative Group for International Agricultural Research |
| NASA | National Aeronautics and Space Administration |
| GSMaP | Global Satellite Mapping of Precipitation |
| PERSIANN-CDR | Precipitation Estimation from Remotely Sensed Information using Artificial Neural Networks |
| JAXA | Japan Aerospace Exploration Agency |
| GLDAS NOAH | Global Land Data Assimilation System Noah Land Surface Model |
| NDVI | Normalized Difference Vegetation Index |
| IMERG | Integrated Multi-satellitE Retrievals for GPM |
| CMA-HD | High-density automatic rain gauge stations across Chinese mainland |
| NEC | North East China |
| NC | North China |
| SCC | South and Central China |
| IM | Inner Mongolia |
| NWC | North West China |
| SWC | South West China |
| QT | Qinghai-Tibet Plateau |
| CDD | Correlation decay distance |
| ADW | Adaptive distance weighting |

| | |
|---|---|
| GBDT | Gradient-boosted decision tree |
| AE | Absolute error |
| KGE | Kling-Gupta efficiency |
| RSD | Relative standard deviation |
| HSS | Heidke skill score |
| FAR | False alarm ratio |
| POD | Probability of detection |

Thank you again for highlighting this issue, which has helped us make the manuscript more readable.

5. In CHM_PRE v1.0, the authors used ADW (Anisotropic Distance Weighting) interpolation, but in v2.0, they reverted to IDW (Inverse Distance Weighting). The rationale for this change is not explained. The authors should clarify why IDW was chosen for v2.0 and how it compares to ADW in terms of performance.

**Response**: Thank you for your valuable comment. In fact, regarding the process of interpolating gauge observations to generate gridded precipitation, the core method used in both CHM_PRE V2 and V1 is the same — an inverse distance weighting (IDW) method with a correlation decay distance (CDD). Based on previous studies (Han et al., 2023; Shen et al., 2010; Xie et al., 2007) and our extensive testing, this method is capable of generating high-accuracy gridded precipitation datasets. In this study, there are three main differences in the interpolation of gauge precipitation compared with CHM_PRE V1:

(1) CHM_PRE V2 restricts the interpolation to the nearest 10 stations when more than 10 stations are available within CDD1, in order to reduce the overestimation of precipitation events in densely gauged areas in eastern China.

(2) In CHM_PRE V1, interpolation was performed by interpolating the ratio of a station's daily precipitation to its daily climatology and then multiplying by the daily climatology. However, for some stations with very low precipitation, this approach could produce extremely large ratios, resulting in unrealistic high precipitation in arid regions. CHM_PRE V2 interpolates the anomalies relative to the climatology instead (He et al., 2020; Zhang et al., 2025) to address this issue.

(3) CHM_PRE V1 used the parameter-elevation regression on independent slopes model (PRISM) climatology data (Daly et al., 1994; Daly et al., 2002) to account for local topographic effects. In contrast, CHM_PRE V2 incorporates local topographic influences such as elevation and slope through a data-driven modeling approach (Section 3.3). Therefore, daily and monthly gridded climatologies were directly calculated and interpolated from gauge observations.

We hope to introduce the production process of the CHM_PRE V2 dataset to its users in a concise manner. After careful consideration, we have rewritten parts of the introduction to better highlight the necessity and significance of upgrading CHM_PRE V1 to V2. However, in the interpolation section based on gauge observations (Section 3.2), we have maintained the

original structure without adding detailed comparisons with CHM_PRE V1, in order to avoid overburdening the readers. The revisions made to the introduction are as follows:

"Our previous study developed a gridded precipitation dataset for the Chinese mainland (a member of the China Hydro-Meteorology datasets, hereinafter called CHM_PRE V1) based on inverse-distance weighting interpolation method and parameter-elevation regression on independent slopes model (PRISM) (Daly et al., 1994, 2002), using data from 2,839 gauges. The CHM_PRE V1 demonstrates overall high accuracy across the Chinese mainland (Han et al., 2023), and has received widespread attention and extensive use, benefiting a large number of hydro-meteorological related studies (Hu et al., 2024; Wan and Zhou, 2024; Yin et al., 2025). However, interpolation-based precipitation datasets rely heavily on ground meteorological gauges, performing poorly in areas with sparse station distribution or missing data.

In summary, a key limitation of existing datasets is that they tend to focus on either spatial autocorrelation or a limited set of precipitation-related covariates, but rarely incorporate multiple types of information simultaneously. However, precipitation is influenced not only by spatial autocorrelation—that is, the dependence of precipitation at a given location on surrounding areas (Chen et al., 2010, 2016; Fan et al., 2021; Huff and Shipp, 1969; Tang et al., 2020)—but also by a wide array of covariates, such as elevation, land surface conditions, atmospheric parameters, and recent precipitation events (Adler et al., 2008; Ham et al., 2023; Ravuri et al., 2021; Trucco et al., 2023). This lack of comprehensive consideration for multiple covariates constrains the accuracy of these datasets, particularly in regions with sparse meteorological stations, such as western China (Jiang et al., 2023). Moreover, existing methods tend to generate excessive minor precipitation, leading to an overestimation of precipitation events, which will have considerable impacts on hydrologic modelling (Dong et al., 2020; Kang et al., 2024; Wei et al., 2022).

To address the aforementioned issues, this study introduces a new high-precision, long-term daily gridded precipitation dataset for the Chinese mainland (a member of the China Hydro-Meteorology datasets, hereinafter called CHM_PRE V2). Building on CHM_PRE V1, CHM_PRE V2 integrates precipitation gauges, remote sensing observations, reanalysis data, and various precipitation-related factors. Through the use of advanced spatial interpolation and machine learning algorithms, our method captures spatial autocorrelation while jointly modelling multiple covariates to enhance precipitation accuracy." (Lines 50–72)

6. The term "CMA-HD" is used but not defined. The authors should provide a clear explanation of what this term refers to.

**Response**: We apologize for this issue. We have added the full form of the abbreviation in the revised manuscript. Additionally, a list of abbreviations has been included in the supplementary materials (**Table S1**) to improve the readability of the manuscript. The corresponding revisions are as follows:

"To further validate the reliability of precipitation data, we obtained daily precipitation observations from 72,901 high-density automatic rain gauge stations across the Chinese mainland (hereafter we refer to it as CMA-HD), provided by the National Meteorological Information Center of CMA (Li et al., 2018)." (Lines 144–146)

**Table S1**. List of abbreviations used throughout this paper.

| Abbreviation | Full Term |
|---|---|
| LGBM | Light gradient boosting machine |
| PRISM | Parameter-elevation regression on independent slopes model |
| IDW | Inverse distance weighting |
| CMA | China Meteorological Administration |
| GHCND | Global Historical Climatology Network-Daily |
| NCDC | National Climatic Data Center |
| NOAA | National Oceanic and Atmospheric Administration |
| SRTM | Shuttle Radar Topography Mission |
| DEM | Digital Elevation Model |
| CGIAR-CSI | Consortium for Spatial Information, Consultative Group for International Agricultural Research |
| NASA | National Aeronautics and Space Administration |
| GSMaP | Global Satellite Mapping of Precipitation |
| PERSIANN-CDR | Precipitation Estimation from Remotely Sensed Information using Artificial Neural Networks |
| JAXA | Japan Aerospace Exploration Agency |
| GLDAS NOAH | Global Land Data Assimilation System Noah Land Surface Model |
| NDVI | Normalized Difference Vegetation Index |
| IMERG | Integrated Multi-satellitE Retrievals for GPM |
| CMA-HD | High-density automatic rain gauge stations across Chinese mainland |
| NEC | North East China |
| NC | North China |
| SCC | South and Central China |
| IM | Inner Mongolia |
| NWC | North West China |
| SWC | South West China |
| QT | Qinghai-Tibet Plateau |
| CDD | Correlation decay distance |
| ADW | Adaptive distance weighting |
| GBDT | Gradient-boosted decision tree |
| AE | Absolute error |
| KGE | Kling-Gupta efficiency |
| RSD | Relative standard deviation |
| HSS | Heidke skill score |
| FAR | False alarm ratio |
| POD | Probability of detection |

7. The authors evaluated the dataset using 63,397 station data points. However, instead of interpolating these station data to 0.1-degree grids using IDW or ADW, they averaged the station values within each 0.1-degree grid for accuracy assessment. The rationale for this approach should be explained, as interpolation might provide a more consistent comparison.

**Response**: Thank you for your comment regarding the accuracy evaluation. We fully agree that interpolating the validation stations can generate gridded data with spatial relationships more consistent with CHM_PRE V2. At the same time, directly comparing station observations with the corresponding grid values introduces the issue of missing accuracy assessments for grid cells without validation stations. However, considering the inevitable uncertainties introduced by interpolation—particularly in regions with sparse station coverage such as western China—we believe that comparing raw station observations with their corresponding grid values yields more reliable accuracy assessments.

Thank you again for your valuable comment. To better clarify this issue, we have added the following explanation to Section 3.4:

"There are two approaches to using station observations to validate the accuracy of gridded precipitation data. The first approach involves interpolating the station data—using methods such as IDW—to generate gridded data at the same spatial resolution as the dataset being validated. This method can produce spatially consistent results with the target gridded dataset. However, as previously mentioned, interpolation methods have some limitations and inevitably introduce interpolation-related uncertainties (McMillan et al., 2018; Wagner et al., 2012). Moreover, the uneven spatial distribution of stations makes the validation results in sparsely monitored areas less reliable. The second approach is to directly compare the station observations with the corresponding grid cell values in the dataset being validated. Although this method only provides validation results for grid cells that contain observation stations, it avoids the uncertainties introduced by interpolation and ensures the reliability of the accuracy assessment. In this study, we adopted the second approach for the validation. To align with the 0.1° gridded precipitation data, station observations were mapped onto a 0.1° grid, and the average precipitation of all stations within each grid cell was regarded as the true precipitation value for that grid cell." (Lines 268–278)

8. The units of variables in Equations 1-8 are not provided. The authors should include the units to ensure clarity and reproducibility.

**Response**: Thank you for bringing this issue to our attention. We have added units for all variables in the revised manuscript. The relevant sentences are as follows:

"where $d(G, P_i)$ represents the distance (km) between grid cell $G$ and gauge station $P_i$, and $p$ is the distance weighting exponent." (Lines 191–192)

"where $y$ and $\hat{y}$ represent the observed precipitation values and the gridded precipitation values (mm/day), respectively; $\mu$ denotes the mean value, $\sigma$ signifies the standard deviation" (Lines 285–286)

9. The relative importance plot in Figure 5 is not well explained. Specifically, it is unclear how the relative importance values were calculated and what "2nd-day prior Prec." and "5th-day prior Prec." represent. Are these cumulative values? A more detailed explanation is needed.

**Response**: Thank you for your valuable comment. The previous analysis of relative importance was based on feature importance derived from the LGBM method by the node splitting, as described at the end of Section 3.4 in the previous manuscript. In this revision, we have re-evaluated the use of feature importance and concluded that it may not be sufficiently reliable for explaining the contributions of different variables to precipitation retrieval. Therefore, we have removed the related content on relative importance (**Figure 5**(c)) in the latest manuscript to ensure the manuscript's rigor. The updated **Figure 5** is as follows:

[Figure]

**Figure 5**. (a) time series of monthly precipitation; (b) multi-year mean monthly precipitation from 2001 to 2020.

We also sincerely apologize for not providing a sufficiently clear explanation of the modeling variables. The variable "2nd-day prior Prec." refers to the daily precipitation two days before the current day—it represents only the value of that specific day, not an accumulated amount over multiple days. In the revised manuscript, we have clarified the meaning of the time-related covariates used in the modeling in Section 3.2. We have also added a comprehensive list of variables used in precipitation retrieval (**Table S3** in the supplementary materials) to better illustrate the modeling details. The corresponding revisions are as follows:

"In addition to spatial and environmental variables, precipitation temporal features were also introduced as covariates. Two types of temporal indicators were constructed: (1) the cumulative precipitation of the current month and year, representing broader-scale precipitation conditions; and (2) daily lagged precipitation values from the previous five days, capturing short-term fluctuations. Each of these five recent days was treated as a separate variable. For example, the variable named "1st-day prior Prec." refers to precipitation one day before the current date, while "5th-day prior Prec." corresponds to five days prior." (Lines 131–135)

**Table S3**. The variables used in the precipitation retrieval.

| Variable Type | Variable Name | Description |
|---|---|---|
| | Lat | Latitude of the grid center |

| | | |
|---|---|---|
| Spatial autocorrelation variables | Lon | Longitude of the grid center |
| | Interp. Prec. | Gridded precipitation based on gauge interpolation |
| Precipitation-related covariates | DEM | Average elevation of the grid |
| | Slope | Average slope of the grid |
| | GLDAS Prec. | Precipitation of the grid from GLDAS |
| | Prec. RS | Satellite-derived precipitation of the grid |
| | GLDAS SM | Soil moisture of the grid from GLDAS |
| | NDVI | NDVI of the grid |
| | Annual Prec. | Annual total precipitation of the grid |
| | Monthly Prec. | Monthly total precipitation of the grid |
| | 1st-day prior Prec. | Daily precipitation one day before the current date |
| | 2nd-day prior Prec. | Daily precipitation two day before the current date |
| | 3rd-day prior Prec. | Daily precipitation three day before the current date |
| | 4th-day prior Prec. | Daily precipitation four day before the current date |
| | 5th-day prior Prec. | Daily precipitation five day before the current date |

10. Figure 6 shows notably high absolute errors in the NC and SCC regions. The authors should discuss the potential reasons for these high errors and whether they are related to regional characteristics or methodological limitations.

**Response**: As you rightly pointed out, both CHM_PRE V2 and V1 exhibit larger absolute errors in regions such as NC, SCC, and QT compared to other regions. This is mainly attributed to the higher precipitation amounts in these regions, which naturally lead to greater absolute errors. To further analyze the error characteristics of CHM_PRE V2 across different regions, we calculated the relative error for each dataset (**Figure R1**(a)) as well as the relative error difference between CHM_PRE V2 and V1 in each region (**Figure R1**(b)). The relative error here is defined as the absolute error of each precipitation event divided by the true daily precipitation value, expressed as a percentage (%).

**Figure R1**(a) shows that CHM_PRE V2 has lower relative errors compared to other datasets. **Figure R1**(b) indicates that CHM_PRE V2 exhibits only minor differences in relative error across different regions and performs better than CHM_PRE V1. Therefore, we conclude that CHM_PRE V2 maintains generally stable error levels across regions.

[Figure]

**Figure R1**. (a) relative error for different precipitation datasets on the testing dataset CMA-HD; (b) relative error difference between CHM_PRE V2 and V1 in each region.

In contrast, accuracy metrics that are unaffected by the magnitude of the variable—such as the Kling-Gupta Efficiency (KGE; **Figure 6**(e)) and the Relative Standard Deviation (RSD; **Figure 6**(f))—demonstrate better regional consistency. The variability of KGE and RSD is relatively higher in SWC and QT, which may be attributed to the sparse distribution of precipitation observation stations and the high spatiotemporal variability of precipitation in these areas (Li et al., 2015; Liu et al., 2019). We have added some discussion to Section 4.2, as follows:

"Specifically, **Figure 6**(d) shows that both the CHM_PRE V2 and V1 datasets exhibit larger absolute errors in regions such as NC, SCC, and QT compared to other areas. This is mainly attributed to the higher precipitation amounts in these regions, which naturally lead to greater absolute errors. In contrast, accuracy metrics that are not affected by the magnitude of the variables, such as KGE (**Figure 6**(e)) and RSD (**Figure 6**(f)), demonstrate better stability across different regions. The KGE and RSD in SWC and QT exhibit relatively greater variability, which could possibly be explained by the sparse distribution of precipitation observation stations and the high spatiotemporal variability of precipitation in these regions (Li et al., 2015; Liu et al., 2019)." (Lines 335–341)

[Figure]

**Figure 6**. Accuracy of different precipitation datasets on the testing dataset CMA-HD. The green and yellow boxes in subfigures (d-f) represent CHM_PRE V2 and CHM_PRE V1, respectively. The ideal values for absolute error, KGE, and RSD are 0 mm/day, 1.0, and 1.0, respectively.

11. From the perspective of RSD (Relative Standard Deviation), it appears that GSMaP might have higher accuracy than CHM_PRE v1.0. The authors should address this observation and discuss how CHM_PRE v2.0 compares to GSMaP in terms of performance.

**Response**: Thank you for pointing out this issue. We carefully examined the RSD values across different precipitation datasets and found that CHM_PRE V2 achieved the best performance among all datasets (RSD takes values in the range $(0, +\infty)$, and the optimal value is 1). As shown in **Figure 6**(c), CHM_PRE V2 demonstrates superior accuracy in terms of RSD, with the median RSD across stations being closer to 1.0 compared to other datasets.

Also, we calculated the overall accuracy of precipitation values for each dataset (**Table S5**) and found that CHM_PRE V2 attained an overall RSD of 0.88, which is approximately 4.76% better than the second-best dataset, IMERG (RSD = 0.84). In fact, CHM_PRE V2 outperforms all other datasets in terms of overall MAE, KGE, and RSD. The only exception is the Bias metric, where CHM_PRE V2 (1.05) is slightly worse than GSMaP (1.04).

**Table S5**. Precipitation accuracy of different datasets validated by high-density gauge data. The bolded numbers in the column represent the optimal accuracy values for that metric.

| Dataset Name | MAE (mm/day) | KGE | Bias | RSD |
|---|---|---|---|---|
| CHM_PRE V2 | **1.48** | **0.79** | 1.05 | **0.88** |
| CHM_PRE V1 | 1.67 | 0.70 | 1.12 | 0.78 |
| GSMaP | 2.94 | 0.48 | **1.04** | 0.80 |
| IMERG | 3.27 | 0.44 | 1.12 | 0.84 |
| PERSIANN-CDR | 3.70 | 0.29 | 1.12 | 0.70 |
| GLDAS | 3.69 | 0.31 | 1.04 | 0.79 |

To better clarify this point, we have added the following content to the revised manuscript:

"CHM_PRE V2 achieved an overall MAE, KGE, and RSD of 1.48 mm/day, 0.79, and 0.88, respectively, outperforming other datasets by 12.84%, 12.86%, and 4.76% (Table S5 in the supplementary material)." (Lines 328–330)

[Figure]

**Figure 6**. Accuracy of different precipitation datasets on the testing dataset CMA-HD. The green and yellow boxes in subfigures (d-f) represent CHM_PRE V2 and CHM_PRE V1, respectively. The ideal values for absolute error, KGE, and RSD are 0 mm/day, 1.0, and 1.0, respectively.

We are sincerely grateful for your insightful comments, which have significantly enhanced the quality of our manuscript.

[revised manuscript text omitted]

----------------------------------------------- end line-------------------------------------------------------

In order to make the review of our revision more convenient, we have marked all changes using the "Track Changes" function in Microsoft Word, and have uploaded the "tracked changes" version as Supplementary Material.

**A DETAILED LIST OF THE RESPONSES**
**TO REVIEWER #3**

**Anonymous Reviewer #3 comments**
**General comments:**

This study developed a daily precipitation dataset (CHM_PRE_V2) for China spanning 1960–2023, demonstrating significantly improved accuracy compared to previous version. This dataset holds substantial importance for advancing hydrological and climatic research in China, particularly in regions with sparse ground observations. The authors have invested considerable effort, and the work is suitable for publication in ESSD. Below are the review comments to further enhance the manuscript.

**Response**: We greatly appreciate your careful reading of the manuscript, positive comments, and valuable suggestions. Your thoughtful review has enhanced our paper considerably. The manuscript has been revised thoroughly according to your comments, with our point-by-point responses detailed below.

**Specific comments:**

1. Caution in using the term "spatiotemporal and physical correlations." Physical correlations typically encompass: (1) Static factors (e.g., latitude, longitude, elevation, slope, vegetation); (2) Meteorological dynamic factors (e.g., humidity, wind speed, available precipitation amount); (3) Land-atmosphere interactions (e.g., soil moisture, vegetation indices, sea surface temperature anomalies); (4) Cloud and precipitation microphysics (e.g., cloud-top temperature, precipitation particle scattering).

As shown in Figure 5c (Figure 1), two-thirds of the selected "physical factors" are precipitation-related variables from different temporal scales or data sources, which conflicts with conventional understanding. The authors should either revise these selections or provide explicit justification.

Additionally, the contributions of factors listed in Figure 5c—whether they correspond to monthly or daily precipitation—require clarification. The relationships between physical factors and precipitation are strongly time-scale-dependent, especially for discontinuous daily precipitation. The authors should consider tailoring factor selection to specific temporal scales (e.g., climatological vs. daily precipitation) and regional variations.

**Response**: We sincerely appreciate your insightful comments. As you pointed out, the explanation of "spatiotemporal and physical correlations" in the previous manuscript was unclear. Upon careful consideration, we concluded that introducing these definitions is not necessary. Therefore, in the latest manuscript, we have updated "spatial correlation" to "spatial autocorrelation" to more precisely express the dependence of precipitation at a location on surrounding areas (Chen et al., 2010, 2016; Fan et al., 2021; Huff and Shipp, 1969). Meanwhile, "temporal and physical correlations" have been revised to "precipitation-related covariates." We have made corresponding revisions throughout the manuscript wherever correlations were mentioned to ensure the rigor of the manuscript. The major revisions are summarized as follows:

[revised manuscript text omitted]

Thank you again for your thoughtful comments and support, which have helped us significantly improve the rigor of our manuscript.

2. Clarify the data sources for GLDAS 2.0 and 2.1 precipitation. The authors should specify whether GLDAS 2.0 and 2.1 precipitation are reanalysis, remote sensing, or fused products. The term "Data Assimilation precipitation" is imprecise and should be revised in the text.
**Response**: Thank you for pointing out this issue. In the revised manuscript, we have explicitly clarified that GLDSA precipitation refers to reanalysis precipitation, in order to better address this issue. The corresponding revisions have been made to Figure 1 and the relevant text, as follows:

"Precipitation datasets derived from gauge-based interpolation (CHM_PRE V1 and CHM_PRE V2) demonstrate significantly higher accuracy compared to those based on remote sensing (GSMaP, IMERG, and PERSIANN-CDR) and reanalysis (GLDAS), as evidenced by lower absolute error, higher KGE) and RSD (Figure 6(a-c))." (Lines 325–328)

[Figure]

**Figure 1**. The data used for precipitation retrieval.

3. Highlight key improvements of updated data. A table or summary explicitly comparing critical differences between CHM_PRE_V2 and its predecessor (e.g., input data, methodology, validation metrics) is strongly recommended. This will underscore the dataset's advancements and novelty.

**Response**: We fully agree with your suggestion to further highlight the comparison between CHM_PRE V2 and V1. In the revised manuscript, we have added Section 4.4 titled "Improvements compared to the previous CHM_PRE V1 dataset" to emphasize the advancements and novelty of CHM_PRE V2. Similarly, we have also strengthened the comparison with CHM_PRE V1 in the Abstract, Introduction, and other relevant sections. The corresponding revisions are as follows:

"Building upon the improved inverse distance weighting interpolation method used in our previous dataset CHM_PRE V1, we integrated a machine learning algorithm—light gradient

boosting machine (LGBM)—to incorporate precipitation-related covariates in a data-driven manner." (Lines 17–20)

"Our previous study developed a gridded precipitation dataset for the Chinese mainland (a member of the China Hydro-Meteorology datasets, hereinafter called CHM_PRE V1) based on inverse-distance weighting interpolation method and parameter-elevation regression on independent slopes model (PRISM) (Daly et al., 1994, 2002), using data from 2,839 gauges. The CHM_PRE V1 demonstrates overall high accuracy across the Chinese mainland (Han et al., 2023), and has received widespread attention and extensive use, benefiting a large number of hydro-meteorological related studies (Hu et al., 2024; Wan and Zhou, 2024; Yin et al., 2025). However, interpolation-based precipitation datasets rely heavily on ground meteorological gauges, performing poorly in areas with sparse station distribution or missing data." (Lines 50–56)

"**4.4 Improvements compared to the previous CHM_PRE V1 dataset**
CHM_PRE V2 is a continuation and improvement of our previously published CHM_PRE V1. Therefore, we further summarize the differences between CHM_PRE V2 and CHM_PRE V1 in **Table 1**, and highlight the improvements of CHM_PRE V2 over the previous version by using bold font. It can be observed that CHM_PRE V2 shares the same spatiotemporal resolution and coverage with V1 (except for the extended time range up to 2023), mainly to maintain consistency with other datasets in the CHM family (Zhang et al., 2025). The spatial interpolation method used in CHM_PRE V2 is largely consistent with that in V1, but it incorporates precipitation-related covariates in a data-driven manner by integrating the LGBM method. Eleven precipitation-related variables were considered, including topographic features (elevation and slope), satellite-derived precipitation estimates, reanalysis-based precipitation products, soil moisture, NDVI, recent daily precipitation records, and aggregate precipitation metrics. The inclusion of these covariates allows for a better representation of the spatiotemporal variability of precipitation (Gu et al., 2023; Ma et al., 2025), resulting in improved precipitation accuracy (with MAE and KGE reaching 1.48 mm/day and 0.79, representing improvements of approximately 12.84% and 12.86% compared to CHM_PRE V1, respectively). In addition, the capability of detecting precipitation events is a critical indicator of the accuracy of precipitation datasets (Dong et al., 2020; Kang et al., 2024). CHM_PRE V2 applies a two-stage modelling approach to distinguish and correct precipitation events, which reduces overestimation of such precipitation events and improves event detection accuracy (with FAR and HSS reaching 0.24 and 0.68, respectively, reflecting improvements of approximately 54.17% and 17.24% over CHM_PRE V1). Overall, CHM_PRE V2 demonstrates obvious improvements over CHM_PRE V1 and serves as a high-accuracy daily gridded precipitation dataset for the Chinese mainland.

**Table 1**. Comparison between CHM_PRE V2 and CHM_PRE V1.

| Category | Item | CHM_PRE V1 | CHM_PRE V2 |
|---|---|---|---|
| Metadata | Spatial resolution | 0.1° | 0.1° |
| | Temporal resolution | Daily | Daily |

| | | 18°N–54°N, 72°E–136°E | 18°N–54°N, 72°E–136°E |
|---|---|---|---|
| Spatial coverage | | | |
| Time Span | | 1961–2022 | 1960–2023 |
| Method | Spatial autocorrelation considered | ✓ | ✓ |
| | Interpolation method | Improved IDW method | **Improved IDW method** |
| | Precipitation-related covariates | Only PRISM climatology data | **11 precipitation covariates** |
| | Covariate modelling approach | X | **LGBM** |
| | Precipitation event considered | X | ✓ |
| Accuracy of precipitation value | MAE (mm/day) | 1.67 | **1.48** |
| | KGE | 0.70 | **0.79** |
| | RSD | 0.78 | **0.88** |
| Accuracy of precipitation event | HSS | 0.58 | **0.68** |
| | Accuracy score | 0.79 | **0.85** |
| | FAR | 0.37 | **0.24** |

" (Lines 389–409)

Once again, we sincerely thank you for your valuable comments, which helped us better highlight the novelty and significance of this study.

4. Clarify model configurations in Section 3.3. The manuscript states that two models were developed for "precipitation event retrieval" and "precipitation value retrieval." Please clarify whether these models share identical predictors and weighting schemes. Detailed descriptions of input variables, parameters, and training protocols for each model are essential.

**Response**: Thanks for your constructive suggestion, which has helped us improve the clarity of our methodology. Following your suggestion, we have thoroughly rewritten Sections 2.2 and 3.3 to better describe our modeling approach. In addition, we have added Table S3 in the supplementary materials to more clearly present the variables used for precipitation retrieval. The corresponding revisions are as follows:

[revised manuscript text omitted]

Once again, we sincerely thank you for your insightful comments, which have greatly enhanced the quality of our manuscript.

[revised manuscript text omitted]

-------------------------------------------- end line-----------------------------------------------------

In order to make the review of our revision more convenient, we have marked all changes using the "Track Changes" function in Microsoft Word, and have uploaded the "tracked changes" version as Supplementary Material.